# Equiformer: Equivariant Graph Attention Transformer for 3D Atomistic Graphs

## Abstract

3D-related inductive biases like translational invariance and rotational equivariance are indispensable to graph neural networks operating on 3D atomistic graphs such as molecules. Inspired by the success of Transformers in various domains, we study how to incorporate these inductive biases into Transformers. In this paper, we present Equiformer, a graph neural network leveraging the strength of Transformer architectures and incorporating $SE(3)/E(3)$-equivariant features based on irreducible representations (irreps). Irreps features encode equivariant information in channel dimensions without complicating graph structures. The simplicity enables us to directly incorporate them by replacing original operations with equivariant counterparts. Moreover, to better adapt Transformers to 3D graphs, we propose a novel equivariant graph attention, which considers both content and geometric information such as relative position contained in irreps features. To improve expressivity of the attention, we replace dot product attention with multi-layer perceptron attention and include non-linear message passing. We benchmark Equiformer on two quantum properties prediction datasets, QM9 and OC20. For QM9, among models trained with the same data partition, Equiformer achieves best results on 11 out of 12 regression tasks. For OC20, under the setting of training with IS2RE data and optionally IS2RS data, Equiformer improves upon state-of-the-art models.

## 1 Introduction

Machine learned models can accelerate the prediction of quantum properties of atomistic systems like molecules by learning approximations of *ab initio* calculations [29, 87, 37, 25, 4, 10, 49, 76, 69, 54, 51]. In particular, graph neural networks (GNNs) have gained increasing popularity due to their performance. By modeling atomistic systems as graphs, GNNs naturally treat the set-like nature of collections of atoms, encode the interaction between atoms in node features and update the features by passing messages between nodes. One factor contributing to the success of neural networks is the ability to incorporate inductive biases that exploit the symmetry of data. Take convolutional neural networks (CNNs) for 2D images as an example: Patterns in images should be recognized regardless of their positions, which motivates the inductive bias of translational equivariance. As for atomistic graphs, where each atom has its coordinate in 3D Euclidean space, we consider inductive biases related to 3D Euclidean group $E(3)$, which include equivariance to 3D translation, 3D rotation, and inversion. Concretely, some properties like energy of an atomistic system should be constant regardless of how we shift the system; others like force should be rotated accordingly if we rotate the system. To incorporate these inductive biases, equivariant and invariant neural networks have been proposed. The former leverages geometric tensors like vectors for equivariant node features [71, 79, 43, 23, 4, 5, 51], and the latter augments graphs with invariant information such as distances and angles extracted from 3D graphs [63, 26, 25, 48, 67, 42].

Submitted to 36th Conference on Neural Information Processing Systems (NeurIPS 2022). Do not distribute.

A parallel line of research focuses on applying Transformer networks [77] to other domains like computer vision [9, 16, 72] and graph [18, 44, 84, 65] and has demonstrated widespread success. However, as Transformers were developed for sequence data [15, 3, 7], it is crucial to incorporate domain-related inductive biases. For example, Vision Transformer [16] shows that adopting a pure Transformer to image classification cannot generalize well and achieves worse results than CNNs when trained on only ImageNet [60] since it lacks inductive biases like translational invariance. Note that ImageNet contains over 1.28M images and the size is already larger than that of many quantum properties prediction datasets [59, 56, 10]. Therefore, this highlights the necessity of including correct inductive biases when applying Transformers to the domain of 3D atomistic graphs.

In this work, we present **Equiformer**, an equivariant graph neural network utilizing $SE(3)/E(3)$-**equi**variant features built from irreducible representations (irreps) and **equi**variant attention mechanisms to combine the 3D-related inductive bias with the strength of Trans**former**. Irreps features encode equivariant information in channel dimensions without complicating graph structures. The simplicity enables us to directly incorporate them into Transformers through replacing original operations with equivariant counterparts and introducing an additional equivariant operation called tensor product. Moreover, we propose a novel equivariant graph attention, which considers both content and geometric information such as relative position. Equivariant graph attention improves upon typical attention in Transformers by replacing dot product attention with theoretically stronger multi-layer perceptron attention and including non-linear message passing. With these innovations, Equiformer demonstrates the possibility of generalizing Transformers to 3D atomistic graphs and achieves competitive results on two quantum properties prediction datasets, QM9 [59, 56] and OC20 [10]. For QM9, compared to models trained with the same data partition, Equiformer achieves the best results on 11 out of 12 regression tasks. For OC20, under the setting of training with IS2RE data and optionally IS2RS data, Equiformer improves upon state-of-the-art models.

## 2 Related Works

Here, we focus on equivariant neural networks and discuss other works in Sec. B in appendix.

**Equivariant GNNs.** Equivariant neural networks [71, 43, 79, 23, 50, 73, 4, 38, 64, 62, 76, 5, 70, 46, 51] operate on geometric tensors like type-$L$ vectors to achieve equivariance. The central idea is to use functions of geometry built from spherical harmonics and irreps features to achieve 3D rotational and translational equivariance as proposed in Tensor Field Network (TFN) [71], which generalizes 2D counterparts [81, 12, 13] to 3D Euclidean space [71, 79, 43]. Previous works differ in equivariant operations used in their networks. TFN [71] and NequIP [4] use graph convolution with linear messages, with the latter utilizing extra equivariant gate activations [79]. SEGNN [5] introduces non-linear messages [29, 61] for irreps features, and the non-linear messages use the same gate activation and improve upon linear messages. SE(3)-Transformer [23] adopts an equivariant version of dot product (DP) attention [77, 39] with linear messages, and the attention can support vectors of any degree (type) $L$. Subsequent works on equivariant Transformers [70, 46] follow the practice of DP attention and linear messages but use more specialized architectures considering only type-0 and type-1 vectors. The proposed Equiformer incorporates all the advantages through combining MLP attention with non-linear messages and supporting vectors of any type. Compared to TFN [71], NequIP [4], SEGNN [5], and SE(3)-Transformer [23], the proposed combination of MLP attention and non-linear messages is more expressive than pure linear or non-linear messages and pure MLP or dot product attention. Compared to other equivariant Transformers [70, 46], in addition to being more expressive, the proposed attention mechanism can support vectors of higher degrees and involve higher order tensor product interactions, which can lead to better performance [4, 5].

## 3 Background

### 3.1 $E(3)$ Equivariance

Atomistic systems are often described using coordinate systems. For 3D Euclidean space, we can freely choose coordinate systems and change between them via the symmetries of 3D space: 3D translation, rotation and inversion ($\vec{r} \rightarrow -\vec{r}$). The groups of 3D translation, rotation and inversion form Euclidean group $E(3)$, with the first two forming $SE(3)$, the second being $SO(3)$, and the last two forming $O(3)$. The laws of physics are invariant to the choice of coordinate systems and therefore properties of atomistic systems are equivariant, e.g., when we rotate our coordinate system, quantities like energy remain the same while others like force rotate accordingly. Formally, a function $f$ mapping between vector spaces $X$ and $Y$ is equivariant to a group of transformation $G$ if for any

input $x \in X$, output $y \in Y$ and group element $g \in G$, we have $f(D_X(g)x) = D_Y(g)f(x)$, where $D_X(g)$ and $D_Y(g)$ are transformation matrices parametrized by $g$ in $X$ and $Y$.

Incorporating equivariance into neural networks as inductive biases is crucial as this enables generalizing to unseen data in a predictable manner. For example, 2D convolution $f$ is equivariant to the group of 2D translation, and thus, CNNs can identify patterns at any location even if they have never seen the patterns at that specific location before. For 3D atomistic graphs, we consider the group of $E(3)$. Features and learnable functions should be $E(3)$-equivariant to geometric transformation acting on position $\vec{r}$. In this work, following previous works [71, 43, 79] implemented in e3nn [28], we achieve $SE(3)/E(3)$-equivariance by using equivariant features based on vector spaces of irreducible representations and equivariant operations like tensor product for learnable functions.

## 3.2 Irreducible Representations

A group representation [17, 85] defines the transformation matrices $D_X(g)$ of group elements $g$ that act on a vector space $X$. For 3D Euclidean group $E(3)$, two examples of vector spaces with different transformation matrices are scalars and Euclidean vectors in $\mathbb{R}^3$, i.e., vectors change with rotation while scalars do not. To address translation symmetry, we simply operate on relative positions. Below we focus our discussion on $O(3)$. The transformation matrices of rotation and inversion are separable and commute, and we first discuss irreducible representations of $SO(3)$.

Any group representation of $SO(3)$ on a given vector space can be decomposed into a concatenation of provably smallest transformation matrices called irreducible representations (irreps). Specifically, for group element $g \in SO(3)$, there are $(2L+1)$-by-$(2L+1)$ irreps matrices $D_L(g)$ called Wigner-D matrices acting on $(2L + 1)$-dimensional vector spaces, where degree $L$ is a non-negative integer. $L$ can be interpreted as an angular frequency and determines how quickly vectors change when rotating coordinate systems. $D_L(g)$ of different $L$ act on independent vector spaces. Vectors transformed by $D_L(g)$ are type-$L$ vectors, with scalars and Euclidean vectors being type-0 and type-1 vectors. It is common to index elements of type-$L$ vectors with an index $m$ called order, where $-L \leq m \leq L$.

The group of inversion $\mathbb{Z}_2$ only has two elements, identity and inversion, and two irreps, even $e$ and odd $o$. Vectors transformed by irrep $e$ do not change sign under inversion while those by irrep $o$ do. We create irreps of $O(3)$ by simply multiplying those of $SO(3)$ and $\mathbb{Z}_2$, and we introduce parity $p$ to type-$L$ vectors to denote how they transform under inversion. Therefore, type-$L$ vectors in $SO(3)$ are extended to type-$(L, p)$ vectors in $O(3)$, where $p$ is $e$ or $o$. In the following, we use type-$L$ vectors for the ease of discussion, but we can generalize to type-$(L, p)$ vectors, unless otherwise stated.

**Irreps Features.** We concatenate multiple type-$L$ vectors to form $SE(3)$-equivariant irreps features. Concretely, irreps feature $f$ has $C_L$ type-$L$ vectors, where $0 \leq L \leq L_{max}$ and $C_L$ is the number of channels for type-$L$ vectors. We index irreps features $f$ by channel $c$, degree $L$, and order $m$ and denote as $f_{c,m}^{(L)}$. Different channels of type-$L$ vectors are parametrized by different weights but are transformed with the same Wigner-D matrix $D_L(g)$. Regular scalar features correspond to including only type-0 vectors. This can generalize to $E(3)$ by including inversion and extending $L$ to $(L, p)$.

**Spherical Harmonics.** Euclidean vectors $\vec{r}$ in $\mathbb{R}^3$ can be projected into type-$L$ vectors $f^{(L)}$ by using spherical harmonics (SH) $Y^{(L)}$: $f^{(L)} = Y^{(L)}(\frac{\vec{r}}{||\vec{r}||})$. SH are $E(3)$-equivariant with $D_L(g)f^{(L)} = Y^{(L)}(\frac{D_1(g)\vec{r}}{||D_1(g)\vec{r}||})$. SH of relative position $\vec{r}_{ij}$ generates the first set of irreps features. Equivariant information propagates to other irreps features through equivariant operations like the tensor product.

## 3.3 Tensor Product

We use tensor products to interact different type-$L$ vectors and first discuss the tensor product for $SO(3)$. The tensor product denoted as $\otimes$ uses Clebsch-Gordan coefficients to combine type-$L_1$ vector $f^{(L_1)}$ and type-$L_2$ vector $g^{(L_2)}$ and produces type-$L_3$ vector $h^{(L_3)}$ as follows:

$$h_{m_3}^{(L_3)} = (f^{(L_1)} \otimes g^{(L_2)})_{m_3} = \sum_{m_1=-L_1}^{L_1} \sum_{m_2=-L_2}^{L_2} C_{(L_1,m_1)(L_2,m_2)}^{(L_3,m_3)} f_{m_1}^{(L_1)} g_{m_2}^{(L_2)} \tag{1}$$

where $m_1$ denotes order and refers to the $m_1$-th element of $f^{(L_1)}$. Clebsch-Gordan coefficients $C_{(L_1,m_1)(L_2,m_2)}^{(L_3,m_3)}$ are non-zero only when $|L_1 - L_2| \leq L_3 \leq |L_1 + L_2|$ and thus restrict output vectors to be of certain types. For efficiency, we discard vectors with $L > L_{max}$, where $L_{max}$ is

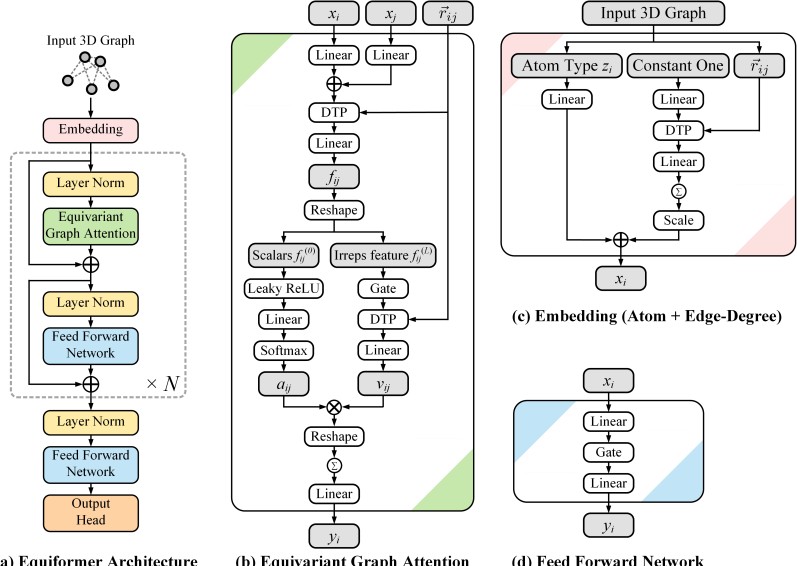

**(a) Equiformer Architecture**  **(b) Equivariant Graph Attention**  **(c) Embedding (Atom + Edge-Degree)**  **(d) Feed Forward Network**

Figure 1: **Architecture of Equiformer.** We embed input 3D graphs with atom and edge-degree embeddings and process them with Transformer blocks, consisting of equivariant graph attention and feed forward networks. In this figure, "$\otimes$" denotes multiplication, "$\oplus$" denotes addition, and "DTP" stands for depth-wise tensor product. $\sum$ within a circle denotes summation over all neighbors. Gray cells indicate intermediate irreps features.

a hyper-parameter, to prevent vectors of increasingly higher dimensions. The tensor product is an equivariant operation, with $(D_{L_1}(g')f^{(L_1)}) \otimes (D_{L_2}(g')g^{(L_2)}) = D_{L_3}(g')h^{(L_3)}$ for $g' \in SO(3)$.

We call each distinct non-trivial combination of $L_1 \otimes L_2 \to L_3$ a path. Each path is independently equivariant, and we can assign one learnable weight to each path in tensor products, which is similar to typical linear layers. We can generalize Eq. 1 to irreps features and include multiple channels of vectors of different types through iterating over all paths associated with channels of vectors. In this way, weights are indexed by $(c_1, l_1, c_2, l_2, c_3, l_3)$, where $c_1$ is the $c_1$-th channel of type-$l_1$ vector in input irreps feature. We use $\otimes_w$ to represent tensor product with weights $w$. Weights can be conditioned on quantities like relative distances. Please refer to Sec. A.4 in appendix for discussion on inversion in tensor products and Sec. D.1 and E.1 for additional results of including inversion.

## 4 Equiformer

We incorporate $SE(3)/E(3)$-equivariant irreps features into Transformers [77] and use equivariant operations. To better adapt Transformers to 3D graph structures, we propose equivariant graph attention. The overall architecture of Equiformer is illustrated in Fig. 1.

### 4.1 Equivariant Operations for Irreps Features

Here we discuss equivariant operations used in Equiformer that serve as building blocks for equivariant graph attention and other modules. They include the equivariant version of the original operations in Transformers and the depth-wise tensor product as illustrated in Fig. 2.

**Linear.**   Linear layers are generalized to irreps features by transforming different type-$L$ vectors separately. Specifically, we apply separate linear operations to each group of type-$L$ vectors. We remove bias terms for non-scalar features with $L > 0$ as biases do not depend on inputs, and therefore, including biases for type-$L$ vectors with $L > 0$ can break equivariance.

**Layer Normalization.**   Transformers adopt layer normalization (LN) [2] to stabilize training. Given input $x \in \mathbb{R}^{N \times C}$, with $N$ being the number of nodes and $C$ the number of channels, LN calculates the linear transformation of normalized input as $\text{LN}(x) = \left( \frac{x - \mu_C}{\sigma_C} \right) \circ \gamma + \beta$, where $\mu_C, \sigma_C \in \mathbb{R}^{N \times 1}$ are mean and standard deviation of input $x$ along the channel dimension, $\gamma, \beta \in \mathbb{R}^{1 \times C}$ are learnable parameters, and $\circ$ denotes element-wise product. By viewing standard deviation as the root mean square value (RMS) of L2-norm of type-$L$ vectors, LN can be generalized to irreps features. Specifically, given input $x \in \mathbb{R}^{N \times C \times (2L+1)}$ of type-$L$ vectors, the output is $\text{LN}(x) =$

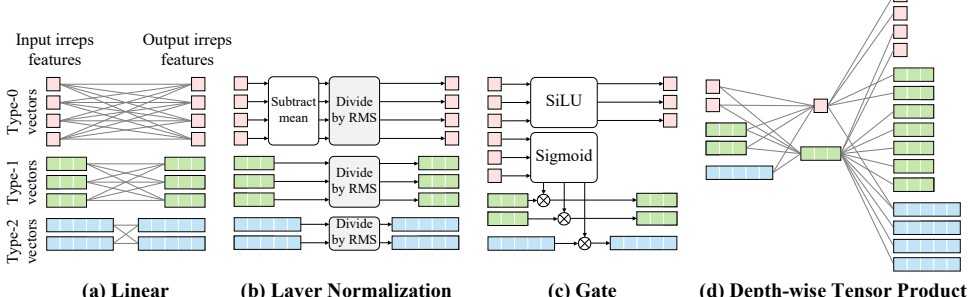

**(a) Linear**   **(b) Layer Normalization**   **(c) Gate**   **(d) Depth-wise Tensor Product**

Figure 2: **Equivariant operations used in Equiformer.** **(a)** Each gray line between input and output irreps features contains one learnable weight. **(b)** "RMS" denotes the root mean square value along the channel dimension. For simplicity, we have removed multiplying by $\gamma$ here. **(c)** Gate layers are equivariant activation functions where non-linearly transformed scalars are used to gate non-scalar irreps features. **(d)** The left two irreps features correspond to two input irreps features, and the rightmost one is the output irreps feature. The two gray lines connecting two vectors in the input irreps features and one vector in the output irreps feature form a path and contain one learnable weight. An alternative visualization of depth-wise tensor products can be found in Fig. 3 in appendix. We only show $SE(3)$-equivariant operations here, and they can be directly generalized to $E(3)$-equivariant features.

170   $\left(\frac{x}{\text{RMS}_C(\text{norm}(x))}\right) \circ \gamma$, where $\text{norm}(x) \in \mathbb{R}^{N \times C \times 1}$ calculates the L2-norm of each type-$L$ vectors in $x$,

171   and $\text{RMS}_C(\text{norm}(x)) \in \mathbb{R}^{N \times 1 \times 1}$ calculates the RMS of L2-norm with mean taken along the channel

172   dimension. We remove mean and biases for type-$L$ vectors with $L \neq 0$ following linear layers.

173   **Gate.**   We use the gate activation [79] for equivariant activation function as shown in Fig. 2(c).

174   Typical activation functions are applied to type-0 vectors. For vectors of higher $L$, we multiply them

175   with non-linearly transformed type-0 vectors for equivariance. Specifically, given input $x$ containing

176   non-scalar $C_L$ type-$L$ vectors with $0 < L \leq L_{max}$ and $(C_0 + \sum_{L=1}^{L_{max}} C_L)$ type-0 vectors, we apply

177   SiLU [19, 55] to the first $C_0$ type-0 vectors and sigmoid function to the other $\sum_{L=1}^{L_{max}} C_L$ type-0

178   vectors to obtain non-linear weights and multiply each type-$L$ vector with corresponding non-linear

179   weights. After the gate activation, the number of channels for type-0 vectors is reduced to $C_0$.

180   **Depth-wise Tensor Product.**   The tensor product defines interaction between vectors of different $L$.

181   To improve its efficiency, we use the depth-wise tensor product (DTP), which restricts one type-$L$

182   vector in output irreps features depends only on one type-$L'$ vector in input irreps features, where $L$

183   can be equal to or different from $L'$. This is similar to depth-wise convolution [34], where one output

184   channel depends on only one input channel. Weights $w$ in the DTP can be input-independent or

185   conditioned on relative distances, and the DTP between two tensors $x$ and $y$ is denoted as $x \otimes_w^{DTP} y$.

## 4.2   Equivariant Graph Attention

187   Self-attention [77, 78, 23, 39, 84, 6] transforms features sent from one spatial location to another

188   with input-dependent weights. We use the notion from Transformers [77] and message passing

189   networks [29, 61, 62, 5] and define message $m_{ij}$ sent from node $j$ to node $i$ as follows:

$$m_{ij} = a_{ij} \times v_{ij} \tag{2}$$

190   where attention weights $a_{ij}$ depend on features on node $i$ and its neighbors $\mathcal{N}(i)$ and values $v_{ij}$

191   are transformed with input-independent weights. In Transformers and Graph Attention Networks

192   (GAT) [78, 6], $v_{ij}$ depends only on node $j$. In message passing networks [29, 61, 62, 5], $v_{ij}$ depends

193   on features on nodes $i$ and $j$ with constant $a_{ij}$. The proposed equivariant graph attention adopts

194   tensor products to incorporate content and geometric information and utilizes multi-layer perceptron

195   attention for $a_{ij}$ and non-linear message passing for $v_{ij}$ as illustrated in Fig. 1(b).

196   **Incorporating Content and Geometric Information.**   Given features $x_i$ and $x_j$ on target node

197   $i$ and source node $j$, we combine the two features with two linear layers to obtain initial message

198   $x_{ij} = \text{Linear}_{dst}(x_i) + \text{Linear}_{src}(x_j)$. $x_{ij}$ is passed to a DTP layer and a linear layer to consider

199   geometric information like relative position contained in different type-$L$ vectors in irreps features:

$$x'_{ij} = x_{ij} \otimes_{w(||\vec{r}_{ij}||)}^{DTP} \text{SH}(\vec{r}_{ij}) \quad \text{and} \quad f_{ij} = \text{Linear}(x'_{ij}) \tag{3}$$

where $x'_{ij}$ is the tensor product of $x_{ij}$ and spherical harmonics embeddings (SH) of relative position $\vec{r}_{ij}$, with weights parametrized by $||\vec{r}_{ij}||$. $f_{ij}$ considers semantic and geometric features on source and target nodes in a linear manner and is used to derive attention weights and non-linear messages.

**Multi-Layer Perceptron Attention.** Attention weights $a_{ij}$ capture how each node interacts with neighboring nodes. $a_{ij}$ are invariant to geometric transformation [23], and therefore, we only use type-0 vectors (scalars) of message $f_{ij}$ denoted as $f_{ij}^{(0)}$ for attention. Note that $f_{ij}^{(0)}$ encodes directional information, as they are generated by tensor products of type-$L$ vectors with $L \geq 0$. Inspired by GATv2 [6], we adopts multi-layer perceptron attention (MLPA) instead of dot product attention (DPA) used in Transformers [77, 39]. In contrast to dot product, MLPs are universal approximators [33, 32, 14] and can theoretically capture any attention patterns [6]. Similar to GAT [78, 6], given $f_{ij}^{(0)}$, we uses one leaky ReLU layer and one linear layer for $a_{ij}$:

$$z_{ij} = a^\top \text{LeakyReLU}(f_{ij}^{(0)}) \quad \text{and} \quad a_{ij} = \text{softmax}_j(z_{ij}) = \frac{\exp(z_{ij})}{\sum_{k \in \mathcal{N}(i)} \exp(z_{ik})} \tag{4}$$

where $a$ is a learnable vectors of the same dimension as $f_{ij}^{(0)}$ and $z_{ij}$ is a single scalar. The output of attention is the sum of value $v_{ij}$ multiplied by corresponding $a_{ij}$ over all neighboring nodes $j \in \mathcal{N}(i)$, where $v_{ij}$ can be obtained by linear or non-linear transformations of $f_{ij}$ as discussed below.

**Non-Linear Message Passing.** Values $v_{ij}$ are features sent from one node to another, transformed with input-independent weights. We first split $f_{ij}$ into $f_{ij}^{(L)}$ and $f_{ij}^{(0)}$, where the former consists of type-$L$ vectors with $0 \leq L \leq L_{max}$ and the latter consists of scalars only. Then, we perform non-linear transformation to $f_{ij}^{(L)}$ to obtain non-linear message:

$$\mu_{ij} = \text{Gate}(f_{ij}^{(L)}) \quad \text{and} \quad v_{ij} = \text{Linear}([\mu_{ij} \otimes_w^{DTP} \text{SH}(\vec{r}_{ij})]) \tag{5}$$

We apply gate activation to $f_{ij}^{(L)}$ to obtain $\mu_{ij}$. We use one DTP and a linear layer to enable interaction between non-linear type-$L$ vectors, which is similar to how we transform $x_{ij}$ into $f_{ij}$. Weights $w$ here are input-independent. We can also use $f_{ij}^{(L)}$ directly as $v_{ij}$ for linear messages.

**Multi-Head Attention.** Following Transformers [77], we can perform $h$ parallel equivariant graph attention functions given $f_{ij}$. The $h$ different outputs are concatenated and projected with a linear layer, resulting in the final output $y_i$ as illustrated in Fig. 1(b). Note that parallelizing attention functions and concatenating can be implemented with "Reshape".

### 4.3 Overall Architecture

For completeness, we discuss other modules in Equiformer here.

**Embedding.** This module consists of atom embedding and edge-degree embedding. For the former, we use a linear layer to transform one-hot encoding of atom species. For the latter, as depicted in the right branch in Fig. 1(c), we first transform a constant one vector into messages encoding local geometry with two linear layers and one intermediate DTP layer and then use sum aggregation to encode degree information [83, 65]. The DTP layer has the same form as that in Eq. 3. We scale the aggregated features by dividing with the squared root of average degrees in training sets so that standard deviation of aggregated features would be close to 1. The two embeddings are summed to produce final embeddings of input 3D graphs.

**Radial Basis and Radial Function.** Relative distances $||\vec{r}_{ij}||$ parametrize weights in some DTP layers. To reflect subtle changes in $||\vec{r}_{ij}||$, we represent distances with Gaussian radial basis with learnable mean and standard deviation [63, 67, 42, 65] or radial Bessel basis [26, 25]. We transform radial basis with a learnable radial function to generate weights for those DTP layers [63, 23, 4]. The function consists of two MLPs with layer normalization [2] and SiLU [19, 55] and a final linear layer.

**Feed Forward Network.** Similar to Transformers, we use two equivariant linear layers and an intermediate gate activation for the feed forward networks in Equiformer.

**Output Head.** The last feed forward network transforms features on each node into a scalar. We perform sum aggregation over all nodes to predict scalar quantities like energy. Similar to edge-degree embedding, we divide the aggregated scalars with the squared root of average numbers of atoms.

| Methods | Task Units | $\alpha$ bohr$^3$ | $\Delta\varepsilon$ meV | $\varepsilon_{\text{HOMO}}$ meV | $\varepsilon_{\text{LUMO}}$ meV | $\mu$ D | $C_\nu$ cal/mol K | $G$ meV | $H$ meV | $R^2$ bohr$^3$ | $U$ meV | $U_0$ meV | ZPVE meV |
|---|---|---|---|---|---|---|---|---|---|---|---|---|---|
| NMP [29] | | .092 | 69 | 43 | 38 | .030 | .040 | 19 | 17 | .180 | 20 | 20 | 1.50 |
| SchNet [63]$^\dagger$ | | .235 | 63 | 41 | 34 | .033 | .033 | 14 | 14 | .073 | 19 | 14 | 1.70 |
| Cormorant [1] | | .085 | 61 | 34 | 38 | .038 | .026 | 20 | 21 | .961 | 21 | 22 | 2.03 |
| LieConv [21] | | .084 | 49 | 30 | 25 | .032 | .038 | 22 | 24 | .800 | 19 | 19 | 2.28 |
| DimeNet++ [25]$^\dagger$ | | .044 | 33 | 25 | 20 | .030 | .023 | 8 | 7 | .331 | 6 | 6 | 1.21 |
| TFN [71]$^\ddagger$ | | .223 | 58 | 40 | 38 | .064 | .101 | - | - | - | - | - | - |
| SE(3)-Transformer [23] | | .142 | 53 | 35 | 33 | .051 | .054 | - | - | - | - | - | - |
| EGNN [62] | | .071 | 48 | 29 | 25 | .029 | .031 | 12 | 12 | .106 | 12 | 11 | 1.55 |
| SphereNet [48]$^\dagger$ | | .046 | 32 | 23 | 18 | .026 | .021 | 8 | 6 | .292 | 7 | 6 | 1.12 |
| SEGNN [5] | | .060 | 42 | 24 | 21 | .023 | .031 | 15 | 16 | .660 | 13 | 15 | 1.62 |
| EQGAT [46] | | .063 | 44 | 26 | 22 | .014 | .027 | 12 | 13 | .257 | 13 | 13 | 1.50 |
| Equiformer | | .056 | 33 | 17 | 16 | .014 | .025 | 10 | 10 | .227 | 11 | 10 | 1.32 |

Table 1: **MAE results on QM9 testing set.** $\dagger$ denotes using different training, validation, testing data partitions as mentioned in SEGNN [5]. $\ddagger$ denotes results from SE(3)-Transformer [23].

| | Energy MAE (eV) $\downarrow$ | | | | | | EwT (%) $\uparrow$ | | | | |
|---|---|---|---|---|---|---|---|---|---|---|---|
| Methods | ID | OOD Ads | OOD Cat | OOD Both | Average | | ID | OOD Ads | OOD Cat | OOD Both | Average |
| SchNet [63]$^\dagger$ | 0.6465 | 0.7074 | 0.6475 | 0.6626 | 0.6660 | | 2.96 | 2.22 | 3.03 | 2.38 | 2.65 |
| DimeNet++ [25]$^\dagger$ | 0.5636 | 0.7127 | 0.5612 | 0.6492 | 0.6217 | | 4.25 | 2.48 | 4.40 | 2.56 | 3.42 |
| GemNet-T [42]$^\dagger$ | 0.5561 | 0.7342 | 0.5659 | 0.6964 | 0.6382 | | 4.51 | 2.24 | 4.37 | 2.38 | 3.38 |
| SphereNet [48] | 0.5632 | 0.6682 | 0.5590 | 0.6190 | 0.6024 | | 4.56 | 2.70 | 4.59 | 2.70 | 3.64 |
| (S)EGNN [5] | 0.5497 | 0.6851 | 0.5519 | 0.6102 | 0.5992 | | 4.99 | 2.50 | 4.71 | 2.88 | 3.77 |
| SEGNN [5] | 0.5310 | 0.6432 | 0.5341 | 0.5777 | 0.5715 | | 5.32 | 2.80 | 4.89 | 3.09 | 4.03 |
| Equiformer | 0.5088 | 0.6271 | 0.5051 | 0.5545 | 0.5489 | | 4.88 | 2.93 | 4.92 | 2.98 | 3.93 |

Table 2: **Results on OC20 IS2RE validation set.** $\dagger$ denotes results reported by SphereNet [48].

| | Energy MAE (eV) $\downarrow$ | | | | | | EwT (%) $\uparrow$ | | | | |
|---|---|---|---|---|---|---|---|---|---|---|---|
| Methods | ID | OOD Ads | OOD Cat | OOD Both | Average | | ID | OOD Ads | OOD Cat | OOD Both | Average |
| CGCNN [82] | 0.6149 | 0.9155 | 0.6219 | 0.8511 | 0.7509 | | 3.40 | 1.93 | 3.10 | 2.00 | 2.61 |
| SchNet [63] | 0.6387 | 0.7342 | 0.6616 | 0.7037 | 0.6846 | | 2.96 | 2.33 | 2.94 | 2.21 | 2.61 |
| DimeNet++ [25] | 0.5621 | 0.7252 | 0.5756 | 0.6613 | 0.6311 | | 4.25 | 2.07 | 4.10 | 2.41 | 3.21 |
| SpinConv [67] | 0.5583 | 0.7230 | 0.5687 | 0.6738 | 0.6310 | | 4.08 | 2.26 | 3.82 | 2.33 | 3.12 |
| SphereNet [48] | 0.5625 | 0.7033 | 0.5708 | 0.6378 | 0.6186 | | 4.47 | 2.29 | 4.09 | 2.41 | 3.32 |
| SEGNN [5] | 0.5327 | 0.6921 | 0.5369 | 0.6790 | 0.6101 | | 5.37 | 2.46 | 4.91 | 2.63 | 3.84 |
| Equiformer | 0.5037 | 0.6881 | 0.5213 | 0.6301 | 0.5858 | | 5.14 | 2.41 | 4.67 | 2.69 | 3.73 |

Table 3: **Results on OC20 IS2RE testing set.**

## 5 Experiment

Our implementation is based on PyTorch [52] (Modified BSD license), PyG [20] (MIT license), e3nn [28] (MIT license), `timm` [80] (Apache-2.0 license), and `ocp` [10] (MIT license).

### 5.1 QM9

**Dataset.** The QM9 [59, 56] dataset (CC BY-NC SA 4.0 license) consisting of 134k small molecules, and the goal is to predict their quantum properties such as energy. We follow the data partition used by Cormorant [1], which has 100k, 18k and 13k molecules in training, validation and testing sets. We minimize mean absolute error (MAE) between prediction and normalized ground truth.

**Setting.** Please refer to Sec. D in appendix for details on architecture and hyper-parameters.

**Result.** We mainly compare with methods trained with the same data partition and summarize the results in Table 1. Equiformer achieves the best results on 11 out of 12 tasks among models trained with same data partition. The comparison to SEGNN [5], which uses irreps features as Equiformer, demonstrates the effectiveness of combining non-lienar messages with MLP attention. Additionally, Equiformer achieves better results for most of tasks when compared to other equivariant Transformers [23, 46], which suggests a better adaption of Transformers to 3D graphs. Besides, the different data partition as denoted by $\dagger$ in Table 1 has 10% more molecules in the training set and less data in the testing set, and this can benefit some tasks that are more dependent on data partitions.

### 5.2 OC20

**Dataset.** The Open Catalyst 2020 (OC20) dataset [10] (Creative Commons Attribution 4.0 License) consists of larger atomic systems, each composed of a small molecule called the adsorbate placed on a large slab called catalyst. The average number of atoms in a system is more than 70, and there are over 50 atom species. The goal is to understand interaction between adsorbates and catalysts through relaxation. An adsorbate is first placed on top of a catalyst to form initial structure (IS). The positions of atoms are updated with forces calculated by density function theory until the system is

| Methods | Energy MAE (eV) ↓ | | | | | EwT (%) ↑ | | | | |
|---|---|---|---|---|---|---|---|---|---|---|
| | ID | OOD Ads | OOD Cat | OOD Both | Average | ID | OOD Ads | OOD Cat | OOD Both | Average |
| GNS [30] | 0.54 | 0.65 | 0.55 | 0.59 | 0.5825 | - | - | - | - | - |
| GNS + Noisy Nodes [30] | 0.47 | 0.51 | 0.48 | 0.46 | 0.4800 | - | - | - | - | - |
| Graphormer [65] | 0.4329 | 0.5850 | 0.4441 | 0.5299 | 0.4980 | - | - | - | - | - |
| Equiformer | 0.4222 | 0.5420 | 0.4231 | 0.4754 | 0.4657 | 7.23 | 3.77 | 7.13 | 4.10 | 5.56 |
| Equiformer + Noisy Nodes | 0.4156 | 0.4976 | 0.4165 | 0.4344 | 0.4410 | 7.47 | 4.64 | 7.19 | 4.84 | 6.04 |

Table 4: **Results on OC20 IS2RE validation set when IS2RS node-level auxiliary task is adopted during training.** "GNS" denotes the 50-layer GNS trained without Noisy Nodes data augmentation, and "GNS + Noisy Nodes" denotes the 100-layer GNS trained with Noisy Nodes. "Equiformer + Noisy Nodes" uses data augmentation of interpolating between initial structure and relaxed structure and adding Gaussian noise as described by Noisy Nodes [30].

| Methods | Energy MAE (eV) ↓ | | | | | EwT (%) ↑ | | | | |
|---|---|---|---|---|---|---|---|---|---|---|
| | ID | OOD Ads | OOD Cat | OOD Both | Average | ID | OOD Ads | OOD Cat | OOD Both | Average |
| GNS + Noisy Nodes [30] | 0.4219 | 0.5678 | 0.4366 | 0.4651 | 0.4728 | 9.12 | 4.25 | 8.01 | 4.64 | 6.5 |
| Graphormer [65]† | 0.3976 | 0.5719 | 0.4166 | 0.5029 | 0.4722 | 8.97 | 3.45 | 8.18 | 3.79 | 6.1 |
| Equiformer + Noisy Nodes | 0.4171 | 0.5479 | 0.4248 | 0.4741 | 0.4660 | 7.71 | 3.70 | 7.15 | 4.07 | 5.66 |

Table 5: **Results on OC20 IS2RE testing set when IS2RS node-level auxiliary task is adopted during training.** † denotes using ensemble of models trained with both IS2RE training and validation sets. In contrast, we use the same single Equiformer model in Table 4, which is trained with only the training set, for evaluation on the testing set.

stable and becomes relaxed structure (RS). The energy of RS, or relaxed energy (RE), is correlated with catalyst activity and therefore a metric for understanding their interaction. We focus on the task of initial structure to relaxed energy (IS2RE), which predicts relaxed energy (RE) given an initial structure (IS). There are 460k, 100k and 100k structures in training, validation, and testing sets, respectively. Performance is measured in MAE and energy within threshold (EwT), the percentage in which predicted energy is within 0.02 eV of ground truth energy. In validation and testing sets, there are four sub-splits containing in-distribution adsorbates and catalysts (ID), out-of-distribution adsorbates (OOD-Ads), out-of-distribution catalysts (OOD-Cat), and out-of-distribution adsorbates and catalysts (OOD-Both).

**Setting.** We consider two training settings based on whether a node-level auxiliary task [30] is adopted. In the first setting, we minimize MAE between predicted energy and ground truth energy without any node-level auxiliary task. In the second setting, we incorporate the task of initial structure to relaxed structure (IS2RS) as a node-level auxiliary task [30]. In addition to predicting energy, we predict node-wise vectors indicating how each atom moves from initial structure to relaxed structure. Please refer to Sec. E in appendix for details on Equiformer architecture and hyper-parameters.

**IS2RE Result without Node-Level Auxiliary Task.** We summarize the results under the first setting in Table 2 and Table 3. Compared with state-of-the-art models like SEGNN [5] and SphereNet [48], Equiformer consistently achieves the lowest MAE for all the four sub-splits in validation and testing sets. Note that energy within threshold (EwT) considers only the percentage of predictions close enough to ground truth and the distribution of errors, and therefore improvement in average errors (MAE) would not necessarily reflect that in error distributions (EwT). Similar phenomena can be observed in Table 3, where for "OOD Both" sub-split, SphereNet [48] achieves lower MAE yet lower EwT than SEGNN [5]. We also note that models in Table 2 and 3 are trained by minimizing MAE and therefore comparing MAE in validation and testing sets could mitigate the discrepancy between training objectives and evaluation metrics and that OC20 leaderboard ranks the relative performance of models mainly according to MAE.

**IS2RE Result with IS2RS Node-Level Auxiliary Task.** We report the results on validation and testing sets under the second setting in Table 4 and Table 5. As of May 20, 2022, Equiformer achieves the best results on IS2RE task when only IS2RE and IS2RS data are used. We note that the proposed Equiformer in Table 5 achieves competitive results even with much less computation. Specifically, training "Equiformer + Noisy Nodes" takes about 24 GPU-days when A6000 GPUs are used. The training time of "GNS + Noisy Nodes" [30] is 56 TPU-days. "Graphormer" [65] uses ensemble of 31 models and requires 372 GPU-days to train all models when A100 GPUs are used. The comparison to GNS demonstrates the improvement from invariant message passing networks to equivariant Transformers. Compared to Graphormer [65], Equiformer demonstrates the effectiveness of equivariant features and the proposed equivariant graph attention. Note that Equiformer, with

| | Methods | | | Task Unit | $\alpha$ bohr$^3$ | $\Delta\varepsilon$ meV | $\varepsilon_{\text{HOMO}}$ meV | $\varepsilon_{\text{LUMO}}$ meV | $\mu$ D | $C_\nu$ cal/mol K |
|---|---|---|---|---|---|---|---|---|---|---|
| Index | Non-linear message passing | MLP attention | Dot product attention | | | | | | | |
| 1 | ✓ | ✓ | | | .056 | 33 | 17 | 16 | .014 | .025 |
| 2 | | ✓ | | | .061 | 34 | 18 | 17 | .015 | .025 |
| 3 | | | ✓ | | .060 | 34 | 18 | 18 | .015 | .026 |

Table 6: **Ablation study results on QM9.**

| | Methods | | | Energy MAE (eV) ↓ | | | | |
|---|---|---|---|---|---|---|---|---|
| Index | Non-linear message passing | MLP attention | Dot product attention | ID | OOD Ads | OOD Cat | OOD Both | Average |
| 1 | ✓ | ✓ | | 0.5088 | 0.6271 | 0.5051 | 0.5545 | 0.5489 |
| 2 | | ✓ | | 0.5168 | 0.6308 | 0.5088 | 0.5657 | 0.5555 |
| 3 | | | ✓ | 0.5386 | 0.6382 | 0.5297 | 0.5692 | 0.5689 |

| | Methods | | | EwT (%) ↑ | | | | |
|---|---|---|---|---|---|---|---|---|
| Index | Non-linear message passing | MLP attention | Dot product attention | ID | OOD Ads | OOD Cat | OOD Both | Average |
| 1 | ✓ | ✓ | | 4.88 | 2.93 | 4.92 | 2.98 | 3.93 |
| 2 | | ✓ | | 4.59 | 2.82 | 4.79 | 3.02 | 3.81 |
| 3 | | | ✓ | 4.37 | 2.60 | 4.36 | 2.86 | 3.55 |

Table 7: **Ablation study results on OC20 IS2RE validation set.**

18 Transformer blocks, is relatively shallow as GNS trained with Noisy Nodes has 100 blocks and Graphormer has 48 Transformer blocks and that deeper networks can typically obtain better results when IS2RS auxiliary task is adopted [30].

## 5.3 Ablation Study

We conduct ablation studies on the improvements brought by MLP attention and non-linear messages in the proposed equivariant graph attention. We modify dot product (DP) attention [77, 23] so that it only differs from MLP attention in how attention weights $a_{ij}$ are generated from $f_{ij}$. Please refer to Sec. C.3 in appendix for details on DP attention. For experiments on QM9 and OC20, unless otherwise stated, we follow the hyper-parameters used in previous experiments.

**Result on QM9.** The comparison is summarized in Table 6. Non-linear messages improve upon linear messages when MLP attention is used. Similar to what is reported by GATv2 [6], the improvement of replacing DP attention with MLP attention is not very significant. We conjecture that DP attention with linear operations is expressive enough to capture common attention patterns as the numbers of nighboring nodes and atom species are much smaller than those in OC20. However, MLP attention is roughly 7% faster as it directly generates scalar features and attention weights from $f_{ij}$ instead of producing additional key and query irreps features for attention weights.

**Result on OC20.** We consider the setting of training without IS2RS auxiliary task and use a smaller learning rate $1.5 \times 10^{-4}$ for DP attention as this improves the performance. We summarize the comparison in Table 7. Non-linear messages consistently improve upon linear messages. In contrast to the results on QM9, MLP attention achieves better performance than DP attention. We surmise this is because OC20 contains larger atomistic graphs with more diverse atom species and therefore requires more expressive attention mechanisms.

## 6 Conclusion and Broader Impact

In this work, we propose Equiformer, a graph neural network (GNN) combining the strengths of Transformers and equivariant features based on irreducible representations (irreps). With irreps features, we build upon existing generic GNNs and Transformer networks [77, 16, 84, 45, 47] by incorporating equivariant operations like tensor products. We further propose equivariant graph attention, which incorporates multi-layer perceptron attention and non-linear messages. Experiments on QM9 and OC20 demonstrate both the effectiveness of Equiformer and the advantage of equivariant graph attention over typical dot product attention.

The broader impact lies in two aspects. First, Equiformer demonstrates the possibility of adapting Transformers to domains such as physics and chemistry, where data can be represented as 3D atomistic graphs. Second, Equiformer achieves more accurate approximations of quantum properties calculation. We believe there is much more to be gained by harnessing these abilities for productive investigation of molecules and materials relevant to application such as energy, electronics, and pharmaceuticals [10], than to be lost by applying these methods for adversarial purposes like creating hazardous chemicals. Additionally, there are still substantial hurdles to go from the identification of a useful or harmful molecule to its large-scale deployment.

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

# Appendix

# A  Additional Mathematical Background

In this section, we provide additional mathematical background on group equivariance helpful for the discussion of the proposed method. Other works [71, 79, 43, 1, 23, 5] also provide similar background. We encourage interested readers to see these works [85, 17] for more in-depth and pedagogical presentations.

## A.1  Group Theory

**Definition of Groups.**  A group is an algebraic structure that consists of a set $G$ and a binary operator $\circ : G \times G \to G$ and is typically denoted as $G$. Groups satisfy the following four axioms:

1. Closure: $g \circ h \in G$ for all $g, h \in G$.

2. Identity: There exists an identity element $e \in G$ such that $g \circ e = e \circ g = g$ for all $g \in G$.

3. Inverse: For each $g \in G$, there exists an inverse element $g^{-1} \in G$ such that $g \circ g^{-1} = g^{-1} \circ g = e$.

4. Associativity: $f \circ g \circ h = (f \circ g) \circ h = f \circ (g \circ h)$ for all $f, g, h \in G$.

In this work, we focus on 3D rotation, translation and inversion. Relevant groups include:

1. The Euclidean group in three dimensions $E(3)$: 3D rotation, translation and inversion.

2. The special Euclidean group in three dimensions $SE(3)$: 3D rotation and translation.

3. The orthogonal group in three dimensions $O(3)$: 3D rotation and inversion.

4. The special orthogonal group in three dimensions $SO(3)$: 3D rotation.

**Group Representations.**  The actions of groups define transformations. Formally, a transformation acting on vector space $X$ parametrized by group element $g \in G$ is an injective function $T_g : X \to X$. A powerful result of group representation theory is that these transformations can be expressed as matrices which act on vector spaces via matrix multiplication. These matrices are called the group representations. Formally, a group representation $D : G \to GL(N)$ is a mapping between a group $G$ and a set of $N \times N$ invertible matrices. The group representation $D(g) : X \to X$ maps an $N$-dimensional vector space $X$ onto itself and satisfies $D(g)D(h) = D(g \circ h)$ for all $g, h \in G$.

How a group is represented depends on the vector space it acts on. If there exists a change of basis $P$ in the form of an $N \times N$ matrix such that $P^{-1}D(g)P = D'(g)$ for all $g \in G$, then we say the two group representations are equivalent. If $D'(g)$ is block diagonal, which means that $g$ acts on independent subspaces of the vector space, the representation $D(g)$ is reducible. A particular class of representations that are convenient for composable functions are irreducible representations or "irreps", which cannot be further reduced. We can express any group representation of $SO(3)$ as a direct sum (concatentation) of irreps [85, 17, 28]:

$$D(g) = P^{-1} \left( \bigoplus_i D_{l_i}(g) \right) P = P^{-1} \begin{pmatrix} D_{l_0}(g) & & \\ & D_{l_1}(g) & \\ & & \cdots \end{pmatrix} P \qquad (6)$$

where $D_{l_i}(g)$ are Wigner-D matrices with degree $l_i$ as metnioned in Sec. 3.2.

## A.2  Equivariance

**Definition of Equivariance and Invariance.**  Equivariance is a property of a function $f : X \to Y$ mapping between vector spaces $X$ and $Y$. Given a group $G$ and group representations $D_X(g)$ and $D_Y(g)$ in input and output spaces $X$ and $Y$, $f$ is equivariant to G if $D_Y(g)f(x) = f(D_X(g)x)$ for all $x \in X$ and $g \in G$. Invariance corresponds to the case where $D_Y(g)$ is the identity $I$ for all $g \in G$.

**Equivariance in Neural Networks.** Group equivariant neural networks are guaranteed to to make equivariant predictions on data transformed by a group. Additionally, they are found to be data-efficient and generalize better than non-symmetry-aware and invariant methods [4, 54, 22]. For 3D atomistic graphs, we consider equivariance to the Euclidean group $E(3)$, which consists of 3D rotation, translation and inversion. For translation, we operate on relative positions and therefore our networks are invariant to 3D translation. We achieve equivariance to rotation and inversion by representing our input data, intermediate features and outputs in vector spaces of $O(3)$ irreps and acting on them with only equivariant operations.

### A.3 Equivariant Features Based on Vector Spaces of Irreducible Representations

**Irreps Features.** As discussed in Sec. 3.2 in the main text, we use type-$L$ vectors for $SE(3)$-equivariant irreps features[1] and type-$(L, p)$ vectors for $E(3)$-equivariant irreps features. Parity $p$ denotes whether vectors change sign under inversion and can be either $e$ (even) or $o$ (odd). Vectors with $p = o$ change sign under inversion while those with $p = e$ do not. Scalar features correspond to type-0 vectors in the case of $SE(3)$-equivariance and correspond to type-$(0, e)$ in the case of $E(3)$-equivariance whereas type-$(0, o)$ vectors correspond to pseudo-scalars. Euclidean vectors in $\mathbb{R}^3$ correspond to type-1 vectors and type-$(1, o)$ vectors whereas type-$(1, e)$ vectors correspond to pseudo-vectors. Note that type-$(L, e)$ vectors and type-$(L, o)$ vectors are considered vectors of different types in equivariant linear layers and layer normalizations.

**Spherical Harmonics.** Euclidean vectors $\vec{r}$ in $\mathbb{R}^3$ can be projected into type-$L$ vectors $f^{(L)}$ by using spherical harmonics $Y^{(L)}$: $f^{(L)} = Y^{(L)}(\frac{\vec{r}}{||\vec{r}||})$ [68]. This is equivalent to the Fourier transform of the angular degree of freedom $\frac{\vec{r}}{||\vec{r}||}$, which can be optionally weighted by $||\vec{r}||$. In the case of $SE(3)$-equivariance, $f^{(L)}$ transforms in the same manner as type-$L$ vectors. For $E(3)$-equivariance, $f^{(L)}$ behaves as type-$(L, p)$ vectors, where $p = e$ if $L$ is even and $p = o$ if $L$ is odd.

**Vectors of Higher $L$ and Other Parities.** Although previously we have restricted concrete examples of vector spaces of $O(3)$ irreps to commonly encountered scalars (type-$(0, e)$ vectors) and Euclidean vectors (type-$(1, o)$ vectors), vector of higher $L$ and other parities are equally physical. For example, the moment of inertia (how an object rotates under torque) transforms as a $3 \times 3$ symmetric matrix, which has symmetric-traceless components behaving as type-$(2, e)$ vectors. Elasticity (how an object deforms under loading) transforms as a rank-4 or $3 \times 3 \times 3 \times 3$ symmetric tensor, which includes components acting as type-$(4, e)$ vectors.

### A.4 Tensor Product

**Tensor Product for $O(3)$.** We use tensor products to interact different type-$(L, p)$ vectors. We extend our discussion in Sec. 3.3 in the main text to include inversion and type-$(L, p)$ vectors. The tensor product denoted as $\otimes$ uses Clebsch-Gordan coefficients to combine type-$(L_1, p_1)$ vector $f^{(L_1, p_1)}$ and type-$(L_2, p_2)$ vector $g^{(L_2, p_2)}$ and produces type-$(L_3, p_3)$ vector $h^{(L_3, p_3)}$ as follows:

$$h_{m_3}^{(L_3, p_3)} = (f^{(L_1, p_1)} \otimes g^{(L_2, p_2)})_{m_3} = \sum_{m_1=-L_1}^{L_1} \sum_{m_2=-L_2}^{L_2} C_{(L_1, m_1)(L_2, m_2)}^{(L_3, m_3)} f_{m_1}^{(L_1, p_1)} g_{m_2}^{(L_2, p_2)} \tag{7}$$

$$p_3 = p_1 \times p_2 \tag{8}$$

The only difference of tensor products for $O(3)$ as described in Eq. 7 from those for $SO(3)$ described in Eq. 1 is that we additionally keep track of the output parity $p_3$ as in Eq. 8 and use the following multiplication rules: $e \times e = e$, $o \times o = e$, and $e \times o = o \times e = o$. For example, the tensor product of a type-$(1, o)$ vector and a type-$(1, e)$ vector can result in one type-$(0, o)$ vector, one type-$(1, o)$ vector, and one type-$(2, o)$ vector.

---

[1]In SEGNN [5], they are also referred to as steerable features. We use the term "irreps features" to remain consistent with `e3nn` [28] library.

**Clebsch-Gordan Coefficients.** The Clebsch-Gordan coefficients for $SO(3)$ are computed from integrals over the basis functions of a given irreducible representation, e.g., the real spherical harmonics, as shown below and are tabulated to avoid unnecessary computation.

$$C^{(L_3, m_3)}_{(L_1, m_1)(L_2, m_2)} = |L_1 m_1; L_2 m_2\rangle \langle L_3 m_3| = \int d\Omega Y^{(L_1)*}_{m_1}(\Omega) Y^{(L_2)*}_{m_2}(\Omega) Y^{(L_3)}_{m_3}(\Omega) \qquad (9)$$

For many combinations of $L_1$, $L_2$, and $L_3$, the Clebsch-Gordan coefficients are zero. The gives rise to the following selection rule for non-trivial coefficients: $-|L_1 + L_2| \le L_3 \le |L_1 + L_2|$.

**Examples of Tensor Products.** Tensor products generally define the interaction between different type-$(L, p)$ vectors in a symmetry-preserving manner and consist of common operations as follows:

1. Scalar-scalar multiplication: scalar $(L = 0, p = e) \otimes$ scalar $(L = 0, p = e) \rightarrow$ scalar $(L = 0, p = e)$.
2. Scalar-vector multiplication: scalar $(L = 0, p = e) \otimes$ vector $(L = 1, p = o) \rightarrow$ vector $(L = 1, p = o)$.
3. Vector dot product: vector $(L = 1, p = o) \otimes$ vector $(L = 1, p = o) \rightarrow$ scalar $(L = 0, p = e)$.
4. Vector cross product: vector $(L = 1, p = o) \otimes$ vector $(L = 1, p = o) \rightarrow$ pseudo-vector $(L = 1, p = e)$.

# B  Related Works

## B.1  Graph Neural Networks for 3D Atomistic Graphs

Graph neural networks (GNNs) are well adapted to perform property prediction of atomic systems because they can handle discrete and topological structures. There are two main ways to represent atomistic graphs [74], which are chemical bond graphs, sometimes denoted as 2D graphs, and 3D spatial graphs. Chemical bond graphs use edges to represent covalent bonds without considering 3D geometry. Due to their similarity to graph structures in other applications, generic GNNs [31, 29, 41, 83, 78, 6] can be directly applied to predict their properties [59, 56, 57, 36, 35]. On the other hand, 3D spatial graphs consider positions of atoms in 3D spaces and therefore 3D geometry. Although 3D graphs can faithfully represent atomistic systems, one challenge of moving from chemical bond graphs to 3D spatial graphs is to remain invariant or equivariant to geometric transformation acting on atom positions. Therefore, invariant neural networks and equivariant neural networks have been proposed for 3D atomistic graphs, with the former leveraging invariant information like distances and angles and the latter operating on geometric tensors like type-$L$ vectors.

## B.2  Invariant GNNs

Previous works [63, 82, 75, 26, 25, 53, 48, 67, 42] extract invariant information from 3D atomistic graphs and operate on the resulting invariant graphs. They mainly differ in leveraging different geometric information such as distances, bond angles (3 atom features) or dihedral angles (4 atom features). SchNet [63] uses relative distances and proposes continuous-filter convolutional layers to learn local interaction between atom pairs. DimeNet series [26, 25] incorporate bond angles by using triplet representations of atoms. SphereNet [48] and GemNet [42, 27] further extend to consider dihedral angles for better performance. In order to consider directional information contained in angles, they rely on triplet or quadruplet representations of atoms. In addition to being memory-intensive [69], they also change graph structures by introducing higher-order interaction terms [11], which would require non-trivial modifications to generic GNNs in order to apply them to 3D graphs. In contrast, the proposed Equiformer uses equivariant irreps features to consider directional information without complicating graph structures and therefore can directly inherit the design of generic GNNs.

## B.3  Attention and Transformer

**Graph Attention.** Graph attention networks (GAT) [78, 6] use multi-layer perceptrons (MLP) to calculate attention weights in a similar manner to message passing networks. Subsequent works

using graph attention mechanisms follow either GAT-like MLP attention [8, 40] or Transformer-like dot product attention [86, 24, 66, 18, 40, 44]. In particular, Kim *et al.* [40] compares these two types of attention mechanisms empirically under a self-supervised setting. Brody *et al.* [6] analyzes their theoretical differences and compares their performance in general settings.

**Graph Transformer.**    A different line of research focuses on adapting standard Transformer networks to graph problems [18, 58, 44, 84, 65]. They adopt dot product attention in Transformers [77] and propose different approaches to incorporate graph-related inductive biases into their networks. GROVE [58] includes additional message passing layers or graph convolutional layers to incorporate local graph structures when calculating attention weights. SAN [44] proposes to learn position embeddings of nodes with full Laplacian spectrum. Graphormer [84] proposes to encode degree information in centrality embeddings and encode distances and edge features in attention biases. The proposed Equiformer belongs to one of these attempts to generalize standard Transformers to graphs and is dedicated to 3D graphs. To incorporate 3D-related inductive biases, we adopt an equivariant version of Transformers with irreps features and propose novel equivariant graph attention.

# C    Details of Architecture

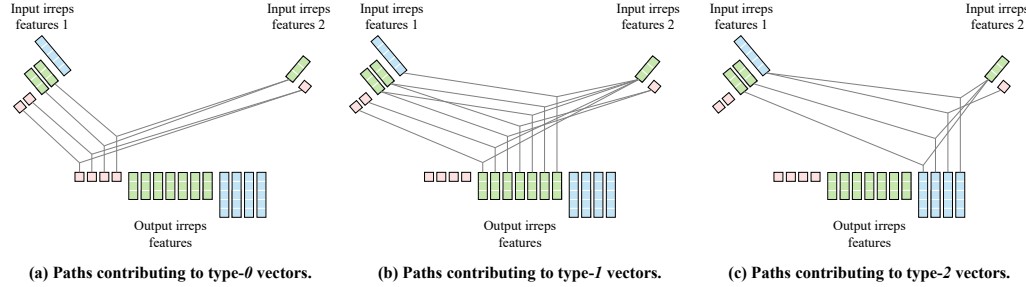

(a) Paths contributing to type-*0* vectors.    (b) Paths contributing to type-*1* vectors.    (c) Paths contributing to type-*2* vectors.

Figure 3: **An alternative visualization of the depth-wise tensor product.**    We follow the visualization of tensor products in `e3nn` [28] and separate paths into three parts based on the types of output vectors. We note that one vector in the output irreps feature depends only on one vector in each input irreps feature.

## C.1    Equivariant Operation Used in Equiformer

We illustrate the equivariant operations used in Equiformer in Fig. 2 and provide an alternative visualization of depth-wise tensor products in Fig. 3.

## C.2    Equiformer Architecture

For simplicity and because most works we compare with do not include equivariance to inversion, we adopt $SE(3)$-equivariant irreps features in Equiformer for experiments in the main text and note that $E(3)$-equivariant irreps features can be easily incorporated into Equiformer.

We define architectural hyper-parameters like the number of channels in some layers in Equiformer, which are used to specify the detailed architectures in Sec. D and Sec. E.

We use $d_{embed}$ to denote embedding dimension, which defines the dimension of most irreps features. Specifically, all irreps features $x_i, y_i$ in Fig. 1 have dimension $d_{embed}$ unless otherwise stated. Besides, we use $d_{sh}$ to represent the dimension of spherical harmonics embeddings of relative positions in all depth-wise tensor products.

For equivariant graph attention in Fig. 1(b), the first two linear layers have the same output dimension $d_{embed}$. The output dimension of depth-wise tensor products (DTP) are determined by that of input irreps features. Equivariant graph attention consists of $h$ parallel attention functions, and the value vector in each attention function has dimension $d_{head}$. We refer to $h$ and $d_{head}$ as the number of heads and head dimension, respectively. By default, we set the number of channels in scalar feature

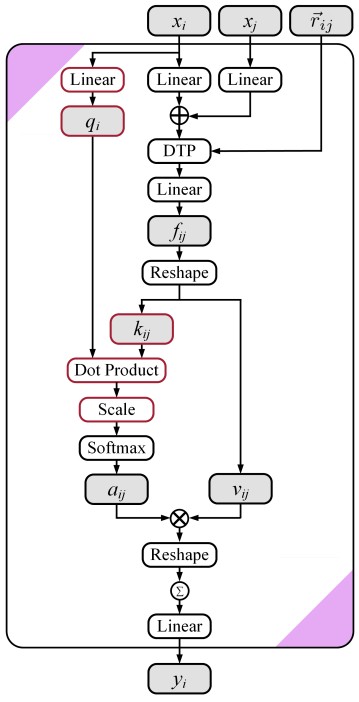

Figure 4: **Architecture of equivariant dot product attention without non-linear message passing.** In this figure, "$\otimes$" denotes multiplication, "$\oplus$" denotes addition, and "DTP" stands for depth-wise tensor product. $\sum$ within a circle denotes summation over all neighbors. Gray cells indicate intermediate irreps features. We highlight the difference of dot product attention from multi-layer perceptron attention in red. Note that key $k_{ij}$ and value $v_{ij}$ are irreps features and therefore $f_{ij}$ in dot product attention typically has more channels than that in multi-layer perceptron attention.

$f_{ij}^{(0)}$ to be the same as the number of channels of type-0 or type-$(0, e)$ vectors in $v_{ij}$. When non-linear messages are adopted in $v_{ij}$, we set the dimension of output irreps features in gate activation to be $h \times d_{head}$. Therefore, we can use two hyper-parameters $h$ and $d_{head}$ to specify the detailed architecture of equivariant graph attention.

As for feed forward networks (FFNs), we denote the dimension of output irreps features in gate activation as $d_{ffn}$. The FFN in the last Transformer block has output dimension $d_{feature}$, and we set $d_{ffn}$ of the last FFN, which is followed by output head, to be $d_{feature}$ as well. Thus, two hyper-parameters $d_{ffn}$ and $d_{feature}$ are used to specify architectures of FFNs and the output dimension after Transformer blocks.

Irreps features contain channels of vectors with degrees up to $L_{max}$. We denote $C_L$ type-$L$ vectors as $(C_L, L)$ and $C_{(L,p)}$ type-$(L, p)$ vectors as $(C_{(L,p)}, L, p)$ and use brackets to represent concatenations of vectors. For example, the dimension of irreps features containing 256 type-0 vectors and 128 type-1 vectors can be represented as $[(256, 0), (128, 1)]$.

## C.3  Dot Product Attention

We illustrate the dot product attention without non-linear message passing used in ablation study in Fig. 4. The architecture is adapted from SE(3)-Transformer [23]. The difference from multi-layer perceptron attention lies in how we obtain attention weights $a_{ij}$ from $f_{ij}$. We split $f_{ij}$ into two irreps features, key $k_{ij}$ and value $v_{ij}$, and obtain query $q_i$ with a linear layer. Then, we perform scaled dot product [77] between $q_i$ and $k_{ij}$ for attention weights.

## C.4  Discussion on Computational Complexity

We discuss the computational complexity of the proposed equivariant graph attention here.

First, we compare dot product attention with MLP attention when linear messages are used for value $v_{ij}$. Dot product attention requires taking the dot product of two irreps features, query $q_i$ and key $k_{ij}$, for attention weights, and both $q_i$ and $k_{ij}$ have the same dimension as value $v_{ij}$. In contrast, MLP attention uses only scalar features $f_{ij}^{(0)}$ for attention weights. The dimension of scalar features $f_{ij}^{(0)}$ is the same as that of the scalar part of $v_{ij}$. Therefore, MLP attention generates less and smaller intermediate features for attention weights and is faster than dot product attention.

Second, compared to linear messages, using non-linear messages increases the number of tensor product operations from 1 to 2. Since tensor products are compute-intensive, this inevitably increases training and inference time.

Please refer to Sec. D.2 and Sec. E.2 for the exact numbers of training time on QM9 and OC20.

# D Details of Experiments on QM9

## D.1 Additional Comparison between $SE(3)$ and $E(3)$ Equivariance

We train two versions of Equiformers, one with $SE(3)$-equivariant features denoted as "Equiformer" and the other with $E(3)$-equivariant features denoted as "$E(3)$-Equiformer", and we compare them in Table 8. Including equivariance to inversion further improves the performance on QM9 dataset.

As for Table 1, we compare "Equiformer" with other works since most of them do not include equivariance to inversion.

| Methods | Task Units | $\alpha$ bohr$^3$ | $\Delta\varepsilon$ meV | $\varepsilon_{\text{HOMO}}$ meV | $\varepsilon_{\text{LUMO}}$ meV | $\mu$ D | $C_\nu$ cal/mol K |
|---|---|---|---|---|---|---|---|
| Equiformer | | .056 | 33 | 17 | 16 | .014 | .025 |
| $E(3)$-Equiformer | | .054 | 32 | 16 | 16 | .013 | .024 |

Table 8: **Ablation study of $SE(3)/E(3)$ equivariance on QM9 testing set.** "Equiformer" operates on $SE(3)$-equivariant features while "$E(3)$-Equiformer" uses $E(3)$-equivariant features. Including inversion further improves mean absolute errors.

## D.2 Training Details

We normalize ground truth by subtracting mean and dividing by standard deviation. For the task of $U$, $U_0$, $G$, and $H$, where single-atom reference values are available, we subtract those reference values from ground truth before normalizing.

We train Equiformer with 6 blocks with $L_{max} = 2$ following SEGNN [5]. We choose Gaussian radial basis [63, 67, 42, 65] for the first six tasks in Table 1 and radial Bessel basis [26, 25] for the others. Table 9 summarizes the hyper-parameters for the QM9 dataset. Further details will be provided in the future. The detailed description of architectural hyper-parameters can be found in Sec. C.2.

We use one A6000 GPU with 48GB to train each model and summarize the computational cost of training for one epoch as follows. Training $E(3)$-Equiformer for one epoch takes about 14.75 minutes. The time of training Equiformer, Equiformer with linear messages (indicated by index 2 in Table 6), and Equiformer with linear messages and dot product attention (indicated by index 3 in Table 6) for one epoch is 11 minutes, 6.6 minutes and 7.1 minutes, respectively.

# E Details of Experiments on OC20

## E.1 Additional Comparison between $SE(3)$ and $E(3)$ Equivariance

We train two versions of Equiformers, one with $SE(3)$-equivariant features denoted as "Equiformer" and the other with $E(3)$-equivariant features denoted as "$E(3)$-Equiformer", and we compare them in Table 10. Including inversion improves the MAE results on ID and OOD Cat sub-splits but degrades the performance on the other sub-splits. Overall, using $E(3)$-equivariant features results in slightly inferior performance. We surmise the reasons are as follows. First, inversion might not be the

| Hyper-parameters | Value or description |
|---|---|
| Optimizer | AdamW |
| Learning rate scheduling | Cosine learning rate with linear warmup |
| Warmup epochs | 5 |
| Maximum learning rate | $5 \times 10^{-4}$ |
| Batch size | 128 |
| Number of epochs | 300 |
| Weight decay | $5 \times 10^{-3}$ |
| Cutoff radius (Å) | 5 |
| Number of radial bases | 128 for Gaussian radial basis, 8 for radial bessel basis |
| Hidden sizes of radial functions | 64 |
| Number of hidden layers in radial functions | 2 |
| | Equiformer |
| Number of Transformer blocks | 6 |
| Embedding dimension $d_{embed}$ | $[(128, 0), (64, 1), (32, 2)]$ |
| Spherical harmonics embedding dimension $d_{sh}$ | $[(1, 0), (1, 1), (1, 2)]$ |
| Number of attention heads $h$ | 4 |
| Attention head dimension $d_{head}$ | $[(32, 0), (16, 1), (8, 2)]$ |
| Hidden dimension in feed forward networks $d_{ffn}$ | $[(384, 0), (192, 1), (96, 2)]$ |
| Output feature dimension $d_{feature}$ | $[(512, 0)]$ |
| | $E(3)$-Equiformer |
| Number of Transformer blocks | 6 |
| Embedding dimension $d_{embed}$ | $[(128, 0, e), (32, 0, o), (32, 1, e), (32, 1, o), (16, 2, e), (16, 2, o)]$ |
| Spherical harmonics embedding dimension $d_{sh}$ | $[(1, 0, e), (1, 1, o), (1, 2, e)]$ |
| Number of attention heads $h$ | 4 |
| Attention head dimension $d_{head}$ | $[(32, 0, e), (8, 0, o), (8, 1, e), (8, 1, o), (4, 2, e), (4, 2, o)]$ |
| Hidden dimension in feed forward networks $d_{ffn}$ | $[(384, 0, e), (96, 0, o), (96, 1, e), (96, 1, o), (48, 2, e), (48, 2, o)]$ |
| Output feature dimension $d_{feature}$ | $[(512, 0, e)]$ |

Table 9: **Hyper-parameters for QM9 dataset.** We denote $C_L$ type-$L$ vectors as $(C_L, L)$ and $C_{(L,p)}$ type-$(L, p)$ vectors as $(C_{(L,p)}, L, p)$ and use brackets to represent concatenations of vectors.

831 key bottleneck. Second, including inversion would break type-1 vectors into two parts, type-$(1, e)$
832 and type-$(1, o)$ vectors. They are regarded as different types in equivariant linear layers and layer
833 normalizations, and therefore, the directional information captured in these two types of vectors can
834 only exchange in depth-wise tensor products. Third, we mainly tune hyper-parameters for Equiformer
835 with $SE(3)$-equivariant features, and it is possible that using $E(3)$-equivariant features would favor
836 different hyper-parameters.

837 For Table 2, 3, 4, and 5, we compare "Equiformer" with other works since most of them do not
838 include equivariance to inversion.

| Methods | Energy MAE (eV) ↓ | | | | | EwT (%) ↑ | | | | |
|---|---|---|---|---|---|---|---|---|---|---|
| | ID | OOD Ads | OOD Cat | OOD Both | Average | ID | OOD Ads | OOD Cat | OOD Both | Average |
| Equiformer | 0.5088 | 0.6271 | 0.5051 | 0.5545 | 0.5489 | 4.88 | 2.93 | 4.92 | 2.98 | 3.93 |
| $E(3)$-Equiformer | 0.5035 | 0.6385 | 0.5034 | 0.5658 | 0.5528 | 5.10 | 2.98 | 5.10 | 3.02 | 4.05 |

Table 10: **Ablation study of $SE(3)$/$E(3)$ equivariance on OC20 IS2RE validation set.**
"Equiformer" operates on $SE(3)$-equivariant features while "$E(3)$-Equiformer" uses $E(3)$-equivariant features.

839 ## E.2 Training Details

840 **IS2RE without Node-Level Auxiliary Task.**  We use hyper-parameters similar to those for QM9
841 dataset and summarize in Table 11. The detailed description of architectural hyper-parameters can be
842 found in Sec. C.2.

843 **IS2RE with IS2RS Node-Level Auxiliary Task.**  We increase the number of Transformer blocks
844 to 18 as deeper networks can benefit more from IS2RS node-level auxiliary task [30]. We follow

the same hyper-parameters in Table 11 except that we increase maximum learning rate to $5 \times 10^{-4}$ and set $d_{feature}$ to $[(512, 0), (256, 1)]$. Inspired by Graphormer [65], we add an extra equivariant graph attention module after the last layer normalization to predict relaxed structures and use a linearly decayed weight for loss associated with IS2RS, which starts at 15 and decays to 1. For Noisy Nodes [30] data augmentation, we first interpolate between initial structure and relaxed structure and then add Gaussian noise as described by Noisy Nodes [30]. When Noisy Nodes data augmentation is used, we increase the number of epochs to 40. Further details will be provided in the future.

We use two A6000 GPUs, each with 48GB, to train models when IS2RS is not included during training. Training Equiformer and $E(3)$-Equiformer takes about $43.6$ and $58.3$ hours. Training Equiformer with linear messages (indicated by index 2 in Table 7) and Equiformer with linear messages and dot product attention (indicated by index 3 in Table 7) takes 30.4 hours and 33.1 hours, respectively. We use four A6000 GPUs to train Equiformer models when IS2RS node-level auxiliary task is adopted during training. Training Equiformer without Noisy Nodes [30] data augmentation takes about 3 days and training with Noisy Nodes takes 6 days. We note that the proposed Equiformer in Table 5 achieves competitive results even with much less computation. Specifically, training "Equiformer + Noisy Nodes" takes about 24 GPU-days when A6000 GPUs are used. The training time of "GNS + Noisy Nodes" [30] is 56 TPU-days. "Graphormer" [65] uses ensemble of 31 models and requires 372 GPU-days to train all models when A100 GPUs are used.

| Hyper-parameters | Value or description |
|---|---|
| Optimizer | AdamW |
| Learning rate scheduling | Cosine learning rate with linear warmup |
| Warmup epochs | 2 |
| Maximum learning rate | $2 \times 10^{-4}$ |
| Batch size | 32 |
| Number of epochs | 20 |
| Weight decay | $1 \times 10^{-3}$ |
| Cutoff radius (Å) | 5 |
| Number of radial basis | 128 |
| Hidden size of radial function | 64 |
| Number of hidden layers in radial function | 2 |
| Equiformer | |
| Number of Transformer blocks | 6 |
| Embedding dimension $d_{embed}$ | $[(256, 0), (128, 1)]$ |
| Spherical harmonics embedding dimension $d_{sh}$ | $[(1, 0), (1, 1)]$ |
| Number of attention heads $h$ | 8 |
| Attention head dimension $d_{head}$ | $[(32, 0), (16, 1)]$ |
| Hidden dimension in feed forward networks $d_{ffn}$ | $[(768, 0), (384, 1)]$ |
| Output feature dimension $d_{feature}$ | $[(512, 0)]$ |
| $E(3)$-Equiformer | |
| Number of Transformer blocks | 6 |
| Embedding dimension $d_{embed}$ | $[(256, 0, e), (64, 0, o), (64, 1, e), (64, 1, o)]$ |
| Spherical harmonics embedding dimension $d_{sh}$ | $[(1, 0, e), (1, 1, o)]$ |
| Number of attention heads $h$ | 8 |
| Attention head dimension $d_{head}$ | $[(32, 0, e), (8, 0, o), (8, 1, e), (8, 1, o)]$ |
| Hidden dimension in feed forward networks $d_{ffn}$ | $[(768, 0, e), (192, 0, o), (192, 1, e), (192, 1, o)]$ |
| Output feature dimension $d_{feature}$ | $[(512, 0, e)]$ |

Table 11: **Hyper-parameters for OC20 dataset under the setting of training without IS2RS auxiliary task.** We denote $C_L$ type-$L$ vectors as $(C_L, L)$ and $C_{(L,p)}$ type-$(L, p)$ vectors as $(C_{(L,p)}, L, p)$ and use brackets to represent concatenations of vectors.

## E.3 Error Distributions

We plot the error distributions of different Equiformer models on different sub-splits of OC20 IS2RE validation set in Fig. 5. For each curve, we sort the absolute errors in ascending order for better visualization and have a few observations. First, for each sub-split, there are always easy examples, for which all models achieve significantly low errors, and hard examples, for which all models have high errors. Second, the performance gains brought by different models are non-uniform among different sub-splits. For example, using MLP attention and non-linear messages improves the errors

on the ID sub-split but is not that helpful on the OOD Ads sub-split. Third, when IS2RS node-level auxiliary task is not included during training, using stronger models mainly improves errors that are beyond the threshold of 0.02 eV, which is used to calculate the metric of energy within threshold (EwT). For instance, on the OOD Both sub-split, using non-linear messages, which corresponds to red and purple curves, improves the absolute errors for the 15000th through 20000th examples. However, the improvement in MAE does not translate to that in EwT as the errors are still higher than the threshold of 0.02 eV. This explains why using non-linear messages in Table 7 improves MAE from $0.5657$ to $0.5545$ but results in almost the same EwT.

## F   Limitations

We discuss several limitations of the proposed Equiformer and equivariant graph attention below.

First, Equiformer is based on irreducible representations (irreps) and therefore can inherit the limitations common to all equivariant networks based on irreps and the library e3nn [28]. For example, using higher degrees $L$ can result in larger features and using tensor products can be compute-intensive. Part of the reasons that tensor products can be computationally expensive are that the kernels have not been heavily optimized and customized as other operations in common libraries like PyTorch [52]. But this is the issue related to software, not the design of networks. While tensor products of irreps naively do not scale well, if all possible interactions and paths are considered, some paths in tensor products can also be pruned for computational efficiency. We leave these potential efficiency gains to future work and in this work focus on general equivariant attention if all possible paths up to $L_{max}$ in tensor products are allowed.

Second, the improvement of the proposed equivariant graph attention can depend on tasks and datasets. For QM9, MLP attention improves not significantly upon dot product attention as shown in Table 6. We surmise that this is because QM9 contains less atoms and less diverse atom types and therefore linear attention is enough. For OC20, MLP attention clearly improves upon dot product attention as shown in Table 7. Non-linear messages improve upon linear ones for the two datasets.

Third, equivariant graph attention requires more computation than typical graph convolution. It includes one softmax operation and thus requires one additional sum aggregation compared to typical message passing. For non-linear message passing, it increases the number of tensor products from one to two and requires more computation. We note that if there is a constraint on training budget, using stronger attention (i.e., MLP attention and non-linear messages) would not always be optimal because for some tasks or datasets, the improvement is not that significant and using stronger attention can slow down training. For example, for the task of $C_\nu$ on QM9, using linear (index 2) or non-linear messages (index 1) results in the same performance as shown in Table 6. However, non-linear messages increase the training time of one epoch from 6.6 minutes to 11 minutes.

Fourth, the proposed attention has complexity proportional to the products of numbers of channels and numbers of edges since the the attention is restricted to local neighborhoods. In the context of 3D atomistic graphs, the complexity is the same as that of messages and graph convolutions. However, in other domains like computer vision, the memory complexity of convolution is proportional to the number of pixels or nodes, not that of edges. Therefore, it would require further modifications in order to use the proposed attention in other domains.

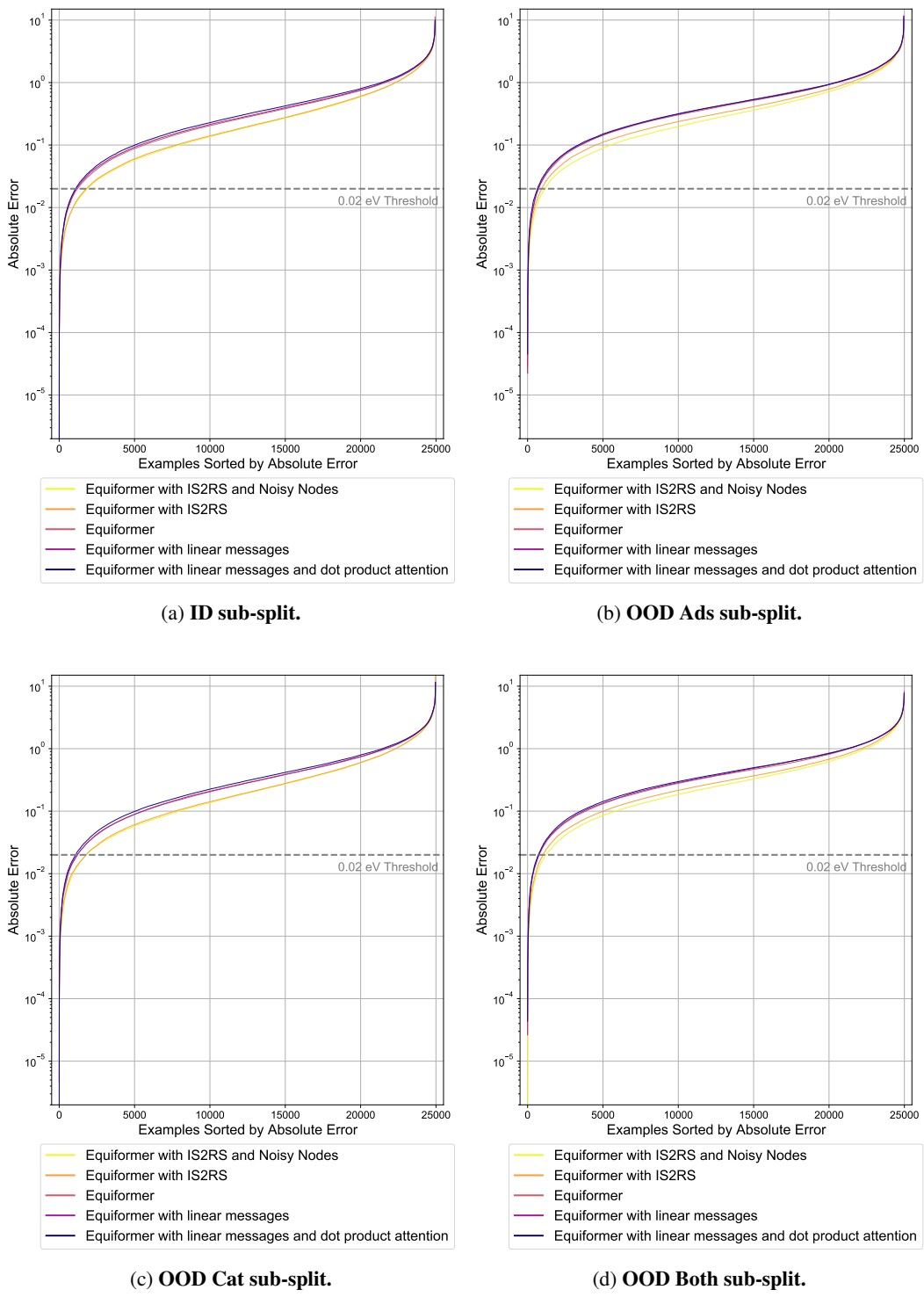

(a) **ID sub-split.**

(b) **OOD Ads sub-split.**

(c) **OOD Cat sub-split.**

(d) **OOD Both sub-split.**

Figure 5: **Error distributions of different Equiformer models on different sub-splits of OC20 IS2RE validation set.**

