# OpenReview forum: "Equiformer: Equivariant Graph Attention Transformer for 3D Atomistic Graphs"
_NeurIPS.cc/2022/Conference — NeurIPS 2022 Submitted_

### Official Review · Reviewer_5yNV · 2022-06-15

**Rating:** 4
**Confidence:** 4
**Soundness:** 2 fair
**Presentation:** 3 good
**Contribution:** 2 fair

**Summary:**

The paper proposes an irreps-based Transformer architecture for atomistic property prediction, named Equiformer. The model leverages irreps to interact between features with different degrees, very much similar to previous literature like TFN, SE(3)-Transformer, or SEGNN. Equiformer further incorporates a GAT-like attention mechanism to enhance the capacity of the model. Equiformer has been experimentally evaluated on two datasets QM9 and the OC20 IS2RE task, obtaining competitive performance against other equivariant graph neural networks.

**Questions:**

The questions below are proposed according to the weaknesses specified in the previous section.

**Q1:**
Are the authors able to systematically discuss the insights on why Equiformer is advantageous over the existing irreps-based methods and equivariant transformer architectures? Potential perspectives would include theoretical analysis (e.g., on the expressivity or universality) or other systematical comparisons and discussions towards this aspect.

**Q2:**
It is strongly recommended to report the existing state-of-the-art results on QM9, like PaiNN and TorchMD-Net (ET). The authors can find these references given in the previous section. By considering these results, the performance of Equiformer on QM9 seems to be not that competitive.

**Q3.**
It would be great to involve running-time comparison/parameter complexity of both Equiformer and the most competitive baseline on both QM9 and OC20. From the current paper, it is very hard to tell where the benefit comes from. One potential concern is related to fairness that Equiformer might be leveraging much higher computational complexity with slower training/inference speed as well as a larger network size.

**Limitations:**

The authors claimed in the checklist that they have discussed the limitations. However, to the best of my effort, I am not able to find them in the paper. The authors are strongly recommended to discuss some limitations, potentially, as specified in Q3, from the high computational complexity of irreps and attention module.

**Strengths And Weaknesses:**

## Strengths
1. Clear presentation with the necessary background on group theory and the irreps-based equivariant construction approach.
2. Detailed depiction of the model architecture and the proposed attention mechanism.
3. Nice ablation studies on both QM9 and OC20 to investigate the role of the proposed message-passing and attention module.

## Weaknesses
1. The technical contribution is somewhat limited given the rich literature on irreps-based equivariant GNNs like TFN [1] and SE(3)-Transformer [2]. There are also equivariant Transformers in this domain, e.g., SE(3)-Transformer [2], TorchMD-Net (ET) [3], and EQGAT [4]. Though some of them have been discussed in the related work, there is still a lack of insightful discussions on why the proposed Equiformer is superior over the so many existing ``Equivariant Transformer'' networks, where the differences lie, and what the key contributing parts are.
2. The enhancement on QM9 is observed to be limited. Some important state-of-the-art models for atomistic property prediction are missing Table 1 (MAE result on QM9 test set), including PaiNN [5] and TorchMD-Net [3]. In particular, TorchMD-Net is also a Transformer-like architecture, which, I believe, is relevant to the paper here and should be compared in Table 1.
3. Lacking discussions about modeling complexity and computational overhead. Generally, the irreps-based methods have already been observed to be quite computationally expensive, compared to other approaches for achieving equivariance, such as the scalar-based networks (e.g., EGNN [6], GMN [7], to name a few). Further incorporating attention will bring extra complexity to the model that is already very heavy to train and inference. There are no comparisons on the running time and number of parameters consumed compared with other baselines.

Refs:

[1] Thomas et al. Tensor field networks: Rotation- and translation-equivariant neural networks for 3D point clouds. 2018.

[2] Fuchs et al. SE(3)-Transformers: 3D Roto-Translation Equivariant Attention Networks. NeurIPS 2020.

[3] Thölke et al. Equivariant Transformers for Neural Network based Molecular Potentials. ICLR 2022.

[4] Le et al. Equivariant graph attention networks for molecular property prediction. 2022.

[5] Schutt et al. Equivariant message passing for the prediction of tensorial properties and molecular spectra. ICML 2021.

[6] Satorras et al. E(n)-equivariant graph neural networks. ICML 2021.

[7] Huang et al. Equivariant Graph Mechanics Networks with Constraints. ICLR 2022.

---

> ### Author Response · Authors · 2022-08-02
> **Response to Reviewer 5yNV (1/4)**
>
> We thank the reviewer 5yNV for the efforts and for acknowledging the __clear presentation of background and the proposed architecture and nice ablation studies to investigate the proposed attention__. We address the comments below.
>
> > 1. [Weakness 1 and Question Q1] Details on comparisons to equivariant networks based on irreducible representations (irreps) and equivariant Transformers.
>
> Please see our general response for detailed comparisons to previous works.
>
> The contributions are:
>
> 1. We find the combination of MLP attention and non-linear message passing improves upon the original dot product attention (Line 12 - Line 14).
>
> 2. We propose the equivariant architecture of MLP attention and non-linear messages (equivariant graph attention). Since features in equivariant networks contain not only scalars but also geometric tensors, the equivariant graph attention requires non-trivial and careful modifications in order to be general and capable of supporting tensors of any degree L.
>
> 3. We show that this equivariant graph attention works well for QM9 and OC20. Particularly, when trained on OC20 IS2RE and IS2RS, the proposed network can improve upon competitive works from the industry. Equiformer achieves lower testing errors and takes 2.33X less training time compared to GNS + Noisy Nodes (ICLR 2022) and 15.5X less training time compared to Graphormer (champion of OC20 challenge in 2021) (Line 392 - Line 395, Table 7 and Table 8 and Line 626 - Line 629 in the appendix).

---

> > ### Author Response · Authors · 2022-08-02
> > **Response to Reviewer 5yNV (2/4)**
> >
> > > 2. [Weakness 2 and Question Q2] Not that significant enhancement on QM9 and comparison to PaiNN and TorchMD-Net.
> >
> > 1. As mentioned in our work (Line 266 - Line 267), we mainly compare models trained with the same data split (e.g., the number of training/validation/testing examples and the index of examples).
> > PaiNN and TorchMD-Net use different data splits and both use 110k examples for training while our work uses 100k training examples. Therefore, directly comparing the results in Table 1 can be unfair due to different training sizes.
> >
> > 2. We additionally train Equiformer with the same data split as TorchMD-Net and summarize the results as below:
> > |                 |   PaiNN   | TorchMD-Net | Equiformer |
> > |-----------------|:---------:|:-----------:|:----------:|
> > | $\mu$             |     0.012 |   **0.011** |  **0.011** |
> > | $\alpha$          | **0.045** |       0.059 |      0.046 |
> > | $\epsilon_{HOMO}$   |      27.6 |        20.3 |     **15** |
> > | $\epsilon_{LUMO}$  |      20.4 |        17.5 |     **14** |
> > | $\delta \epsilon$ |      45.7 |        36.1 |     **30** |
> > | $R^2$             |     0.066 |   **0.033** |      0.251 |
> > | $ZPVE$            |      1.28 |        1.84 |   **1.26** |
> > | $U_0$             |  **5.85** |        6.15 |         10 |
> > | $U$               |  **5.83** |        6.38 |         11 |
> > | $H$               |  **5.98** |        6.16 |         10 |
> > | $G$               |  **7.35** |        7.62 |         11 |
> > | $C_\nu$             |     0.024 |       0.026 |  **0.023** |
> >
> > 3. We note that __Equiformer achieves the best results on 6 out of 12 regression tasks__ and that for the task of $R^2$, PaiNN and TorchMD-Net use specialized architecture that takes into account the prior knowledge of the task and the comparison of this task is less fair. For the task of $U_0$, $U$, $H$ and $G$, we surmise that the training epochs are not enough for Equiformer to converge (the number of epochs is 300, which is 10X less than that of TorchMD-Net). Further tuning hyper-parameters can achieve better results, and we will investigate this. Compared to PaiNN, Equiformer achieves better results on 6 out of 12 regression tasks. Compared to TorchMD-Net, Equiformer achieves better results on 6 tasks, equal results on 1 task and worse results on 5 tasks. Overall, __Equiformer is still competitive to PaiNN and TorchMD-Net on QM9.__
> >
> > 4. Moreover, the improvement of the proposed attention depends on datasets. For QM9, we already showed that the improvement of replacing the typical dot production attention used in previous works of equivariant Transformers with theoretically stronger MLP attention is not very significant and discussed this in the main text (Table 5 and Line 321 - Line 324). For OC20, however, MLP attention clearly improves dot product attention (Table 6). This would suggest that on OC20, Equiformer with MLP attention could improve upon TorchMD-Net with dot product attention.
> >
> > 5. Besides, we also report our results on MD17 and compare Equiformer with PaiNN [5] and TorchMD-Net [3]. __The comparison on MD17 dataset shows that Equiformer clearly improves upon both PaiNN and TorchMD-Net.__ Please see results in the general response.
> >
> > 6. We additionally __compare Equiformer with PaiNN on OC20 IS2RE testing set__ and use the results of PaiNN reported by OC20 team. The results of energy MAE (eV) are summarized below, and __the improvement of Equiformer becomes significant when the dataset contains more atoms and more diverse atom types__.
> >
> > |   Method   |     ID     |   OOD Ads  |  OOD Cat   |  OOD Both  |   Average  |
> > |:----------:|:----------:|:----------:|:----------:|:----------:|:----------:|
> > | PaiNN      |      0.575 |      0.783 |      0.604 |      0.743 |      0.676 |
> > | Equiformer | **0.5037** | **0.6881** | **0.5213** | **0.6301** | **0.5858** |
> >
> >
> > __In summary, Equiformer achieves overall better results on QM9, MD17 and OC20__.
> >
> > 1. Comparison to PaiNN:
> > * (a) QM9: comparable performance with 6 better and 6 worse results.
> > * (b) MD17: Equiformer achieves better results for all 7 molecules.
> > * (c) OC20: Equiformer achieves better results for all sub-splits.
> >
> >
> > 2. Comparison to TorchMD-Net:
> > * (a) QM9: comparable performance with 6 better, 1 equal and 5 worse results.
> > * (b) MD17: (For the first 4 molecules, Equiformer achieves better results. For the last 4 molecules, Equiformer achieves slightly higher energy errors but significantly lower force errors.) After tuning the ratio of force loss and energy loss, Equiformer achieves better results for all molecules.
> > * (c) OC20: Both MLP attention and non-linear messages in Equiformer improve upon dot product attention and linear messages, which is used by TorchMD-Net (Table 6).
> >
> > 3. Additional comparison to SEGNN:
> > * (a) QM9: Equiformer achieves better results for all 12 tasks (Table 1).
> > * (b) OC20: Equiformer achieves better results for all sub-splits (Table 2 and Table 3). Note that SEGNN achieves the second best results on OC20 and better results than PaiNN.

---

> > > ### Author Response · Authors · 2022-08-02
> > > **Response to Reviewer 5yNV (3/4)**
> > >
> > > > 3. [Weakness 3 and Question Q3] Discussion about modeling complexity and computational overhead.
> > >
> > >
> > > Please see our general response for training time (computational overhead).
> > >
> > > We also discussed why MLP attention is faster than dot product attention in our case (Line 325 - Line 326 in the main text). For the modeling complexity and computational overhead, we will add discussion on how and why MLP attention and non-linear message passing affect training time.
> > >
> > > > 4. [Weakness 3 and Question Q3] Comparison of running time and numbers of parameters with the most competitive baselines __SEGNN__ on __QM9__ and __OC20__.
> > >
> > > We report the training time and numbers of parameters and summarize the results below.
> > > |                                                 | QM9                  |               | OC20                 |               |
> > > |-------------------------------------------------|----------------------|---------------|----------------------|---------------|
> > > |                                                 | Number of parameters | Training time | Number of parameters | Training time |
> > > | SEGNN                                           |                1.03M |  81 GPU-hours |                4.21M |  79 GPU-hours |
> > > | Equiformer (MLP attention + non-linear message) |                3.53M |  55 GPU-hours |                9.12M |  87 GPU-hours |
> > > | Equiformer   (MLP attention + linear message)   |                3.01M |  33 GPU-hours |                7.84M |  61 GPU-hours |
> > >
> > > SEGNN is written with the same e3nn library and the comparison is fair.
> > > __Equiformer with MLP attention and non-linear message is faster than SEGNN on QM9 and has similar training time on OC20.__
> > >
> > > Although we use more channels and more parameters, the training time is comparable. The reasons are:
> > >
> > > 1. We use more efficient depth-wise tensor products (DTP), where one output channel depends on only one input channel. SEGNN uses more compute-intensive fully connected tensor products (FCTP), where one output channel depends on all input channels.
> > >
> > > 2. SEGNN uses 4 FCTPs in each message passing block while Equiformer uses only 2 DTP in each block.
> > >
> > > Moreover, we note that Equiformer with MLP attention and linear messages also improves upon SEGNN (compare index 2 in Table 5 with SEGNN in Table 1 and index 2 in Table 6 with SEGNN in Table 2) and is more compute-efficient. Therefore, using only MLP attention in Equiformer improves both performance and training efficiency of using non-linear messages in SEGNN on QM9 and OC20.

---

> > > > ### Author Response · Authors · 2022-08-02
> > > > **Response to Reviewer 5yNV (4/4)**
> > > >
> > > >
> > > > > 5. [Weakness 3 and Question Q3] Comparison of running time and numbers of parameters with the most competitive baselines __TorchMD-Net__ on __QM9__ and __MD17__.
> > > >
> > > > We directly use the code from TorchMD-Net to estimate the training time and summarize the results below.
> > > > |             | QM9                  |                            | MD17                 |                            |
> > > > |-------------|----------------------|----------------------------|----------------------|----------------------------|
> > > > |             | Number of parameters | Training time              | Number of parameters | Training time              |
> > > > | TorchMD-Net |                 6.9M | 92 GPU-hours (3000 epochs) |                 1.3M | 10 GPU-hours (3000 epochs) |
> > > > | Equiformer  |                3.53M |  60 GPU-hours (300 epochs) |                3.53M | 23 GPU-hours (1500 epochs) |
> > > >
> > > > Equiformer takes more training time per epoch, and this is because:
> > > > 1. Equiformer uses more expressive non-linear messages, which compared to linear messages used in other equivariant Transformers, doubles the number of tensor products and therefore almost doubles the training time.
> > > > 2. Equiformer incorporates tensors of higher degrees L (e.g., L=2), which improves performance but slows down the training.
> > > >
> > > > However, we note that:
> > > > 1. On QM9, Equiformer achieves 6 better, 1 equal and 5 worse results and takes 35% less training time.
> > > > 2. On MD17, Equiformer achieves overall better results for all molecules.
> > > >
> > > > Additionally, we would like to emphasize:
> > > > 1. The proposed MLP attention is faster than dot product attention in this context (Line 325 - Line 326).
> > > >
> > > > 2. __Equiformer achieves better performance but slower training time. This does not simply imply other works can spend more computation and get better performance.__ For example, if the performance gain lies in using higher degrees L (e.g., L = 2), TorchMD-Net cannot improve performance as it can use L up to 1. Moreover, non-linear messages slow down training, but linear messages cannot be as expressive as non-linear ones regardless of how much computation is involved or how many channels (larger networks) are used. Non-linear MLP attention and linear dot product attention are of the same case.
> > > >
> > > > 3. __For some tasks, using larger networks (e.g., greater depths) does not translate to better performance.__ As mentioned by Noisy Nodes [1], for OC20 IS2RE when IS2RS is not adopted, using more layers results in the same error. We confirm with this by training Equiformer with 6 and 8 Transformer blocks and observe no difference between the two.
> > > >
> > > > 4. __We do not tune the training efficiency for individual cases but instead focus on generally better architectures.__ For example, as shown in Table 5, using non-linear message does not improve the performance of $C_\nu$. Thus, for this case, we can use a weaker model and save training time.
> > > >
> > > >
> > > > Reference:
> > > >
> > > > [1] Godwin et al. Simple GNN Regularisation for 3D Molecular Property Prediction & Beyond. ICLR 2022.
> > > >
> > > >
> > > > > 6.  Limitations.
> > > >
> > > > We thank the reviewer for pointing out this and for mentioning the computational complexity. Please see our general response for details.

---

> > > > > ### Comment · Reviewer_5yNV · 2022-08-09
> > > > > **Thanks for the response**
> > > > >
> > > > > Thank the authors for providing the detailed response. With these clarifications and additional experiments, I am now open to increase the score, barring any other concern in the reviewer-meta reviewer discussion period. As for now, I am mainly impressed by the exhaustive and thorough experimental evaluations. I still recommend the authors improve the presentation of the paper, by considering highlighting the differences between your approach and the existing works.

---

> > > > > > ### Author Response · Authors · 2022-08-09
> > > > > > **Response to Reviewer 5yNV**
> > > > > >
> > > > > > We thank the reviewer for the response and for acknowledging the thorough experimental evaluation.
> > > > > > We will improve the presentation of this paper and highlight the differences between existing works and this work.

---

### Official Review · Reviewer_qLqa · 2022-07-11

**Rating:** 5
**Confidence:** 4
**Soundness:** 3 good
**Presentation:** 2 fair
**Contribution:** 3 good

**Summary:**

This paper proposes a novel Equiformer model for DFT-level scalar quantum properties prediction based on 3D conformation of molecules or molecules and catalysts system.
Equiformer has permutation equivariant and SE(3)/E(3) equivariant architecture, a novel transformer-like architecture operating on irreducible representations(Irreps) feature.
This paper proposes a novel Depth-wise Tensor Product method, creatively adopts MLP attention for Irreps and borrows the successful Non-Linear Message from SEGNN as Value in attention mechanism.
Experiments are conducted on QM9 and OC20-IS2RE, exhibiting the effectiveness of Non-Linear Message, MLP attention and overall architecture, achieving SOTA performance in both tasks.


**Questions:**

+ All interactions between different type vectors and different elements within vector depend on DTP with $SH(r_{ij})$, while no experiment shows Self (Depth-wise) Tensor Product or Cross (Depth-wise)  Tensor Product cannot improve the performance.
+ Comparison with Dot Product Attention could be more complete, e.g., key can be obtained from $x_j$ with a linear layer and attention bias can be obtained from the scalar part of $f_{ij}$ with a MLP before Reshape (splitting heads) or learnable radial function(mentioned in line 235-239) encoding $||r_{ij}||$, etc.


**Limitations:**

The authors didn’t describe the limitations of their work.

**Strengths And Weaknesses:**

Strengths:
+ Proposes Novel Depth-wise Tensor Product
+ Adopts MLP attention for Irreps and shows its effectiveness
+ Clear Figure 2 in appendix describes how each component works.
+ Proposed S×SE(3) Equivariant Equiformer achieves SOTA performance on QM9 and OC20-IS2RE

Weaknesses:
- Attention Score component could cooperate with Attention Value component, ablation studies only conducted on MLP attention + Non-Linear Message, MLP attention + Linear Message and Dot Product attention + Linear Message. It seems that is enough to show the effectiveness of each component, but the core contribution of this paper is the combination of MLP attention and Non-Linear Message. Thus it might be necessary to compare MLP attention to Dot Product attention when Non-Linear Message is used.
- No experiment or explanation shows the effectiveness/efficiency of DTP.
- Equiformer is only applied to scalar prediction tasks (except auxiliary task), which only need SE(3) invariance or trivial equivariance(i.e. identity representation). The experiments cannot show the significance of the equivariant nature of Equiformer.
- Equiformer architecture is very complicated, while there isn’t a clear motivation or intuition or explanation or thorough experimentation to support it. If the embedding module and the parameterization of DTP weights are considered, Equiformer could be more complicated.
- Section 3.1(line 154-184) is less informative than Figure 2, while occupies more space and harder to understand. Figure 2 is more important than the verbose background introduction in Section 2(Section 2.3 is good, while other parts could be more compact or omitted). And given line 114 has already said $C_L$ is the number of channels for type-L vectors, line 174-175 says “input x containing $(C_0+\sum_{L=1}^{L_{max}}C_L)$ type-0 vectors” is confusing. It might be better to replace this paragraph with Figure 2(c).

---

> ### Author Response · Authors · 2022-08-02
> **Response to Reviewer qLqa (1/3)**
>
> We thank the reviewer qLqa for the efforts and for acknowledging that __the proposed architecture is novel__ and __achieves competitive results on QM9 and OC20__. We address the comments below.
>
> > 1. Thus it might be necessary to compare MLP attention to Dot Product attention when Non-Linear Message is used.
>
> We conduct ablation study and compare MLP attention and dot product attention when non-linear message is used. We summarize the new results (index 4) below and compare with the original results in Table 5 and Table 6. We can see that __MLP attention is more efficient and achieves equal or better results than dot product when non-linear message is used__.
>
> * QM9
>
> | index | Non-linear  message | MLP  attention | Dot product  attention | $\alpha$ | $\Delta \epsilon$ | $\epsilon_{HOMO}$ | $\epsilon_{LUMO}$ | $ \mu$ |  $C_\nu$ |
> |:-----:|:-------------------:|:--------------:|:----------------------:|:------:|:----------------:|:-------------:|:--------------:|:----:|:----:|
> |   1   |          Y          |        Y       |                        |   .056 |               33 |            17 |             16 | .014 | .025 |
> |   2   |                     |        Y       |                        |   .061 |               34 |            18 |             17 | .015 | .025 |
> |   3   |                     |                |            Y           |   .060 |               34 |            18 |             18 | .015 | .026 |
> |   4   |          Y          |                |            Y           |   .056 |               33 |            17 |             16 | .014 | .025 |
>
> For QM9, when non-linear message is used, MLP attention performs on par with dot product attention. This is somewhat expected as non-linear message enables non-linear edge features and that dot product attention is already able to capture attention patterns of relatively small dataset. However, we note that __when non-linear message is used, MLP attention is faster than dot product attention by about 39%__.
>
> * OC20
>
> OC20 energy MAE (eV)
>
> | index | Non-linear  message | MLP  attention | Dot product  attention |   ID   | OOD Ads | OOD Cat | OOD Both | Average |
> |:-----:|:-------------------:|:--------------:|:----------------------:|:------:|:-------:|:-------:|----------|---------|
> |   1   |          Y          |        Y       |                        | 0.5088 |  0.6271 |  0.5051 |   0.5545 |  0.5489 |
> |   2   |                     |        Y       |                        | 0.5168 |  0.6308 |  0.5088 |   0.5657 |  0.5555 |
> |   3   |                     |                |            Y           | 0.5386 |  0.6382 |  0.5297 |   0.5692 |  0.5689 |
> |   4   |          Y          |                |            Y           | 0.5197 |  0.6289 |  0.5149 |   0.5520 |  0.5534 |
>
>
> OC20 EwT (%)
>
> | index | Non-linear  message | MLP  attention | Dot product  attention |  ID  | OOD Ads | OOD Cat | OOD Both | Average |
> |:-----:|:-------------------:|:--------------:|:----------------------:|:----:|:-------:|:-------:|----------|---------|
> |   1   |          Y          |        Y       |                        | 4.88 |    2.93 |    4.92 |     2.98 |    3.93 |
> |   2   |                     |        Y       |                        | 4.59 |    2.82 |    4.79 |     3.02 |    3.81 |
> |   3   |                     |                |            Y           | 4.37 |    2.60 |    4.36 |     2.86 |    3.55 |
> |   4   |          Y          |                |            Y           | 4.45 |    2.85 |    4.43 |     2.94 |    3.67 |
>
> For OC20, we can still observe that MLP attention performs better than dot product attention. The only exception is that on OOD Both, dot product attention performs slightly better. However, for average results, __MLP attention improves dot product attention when non-linear message is used__. We note that __dot product attention + non-linear message (index 4) takes about 56 hours and therefore is about 27% slower than MLP attention + non-linear message (index 1) (44 hours) and 86% slower than MLP attention (index 2) (30 hours)__.

---

> > ### Author Response · Authors · 2022-08-02
> > **Response to Reviewer qLqa (2/3)**
> >
> > > 2. No experiment or explanation shows the effectiveness/efficiency of DTP.
> >
> > Thanks a lot for pointing out this.
> >
> > Here is the explanation and motivation of using depth-wise tensor product (DTP).
> >
> > If we use a fully connected tensor product (FCTP), where each output channel depends on all input channels, the number of weights will be proportional to $C_{in} \times C_{out}$, with $C_{in}$ being the number of input channels and $C_{out}$ the number of output channels. Note that in our network, the weights of tensor products are generated by scalar functions e.g., radial functions (Line 198 - Line 201) and the memory complexity is proportional to the number of weights.
> > If we use FCTP, the memory complexity is $C_{in} \times C_{out}$. If we use DTP instead, the complexity is only $C_{in}$. Thus, using DTP can save memory significantly by $C_{out}$ times and in our experiments, using FCTP instead of DTP can result in out of memory error.
> > Thus, in our network, we can only choose to use DTP.
> >
> >
> > > 3. Equiformer is only applied to scalar prediction tasks (except auxiliary task), which only need SE(3) invariance or trivial equivariance (i.e. identity representation). The experiments cannot show the significance of the equivariant nature of Equiformer.
> >
> > Please see our general response for results on MD17.
> >
> > Besides, Nequip [1] and works by Rackers et al. [2] and Frey et al. [3] have shown that even in cases where the task is to predict invariants, including equivariant features leads to more accurate and generalizable models.
> >
> > Reference:
> >
> > [1] Batzner et al. E(3)-equivariant graph neural networks for data-efficient and accurate interatomic potentials. Nature Communications 2022.
> >
> > [2] Rackers et al. Cracking the quantum scaling limit with machine learned electron densities. https://arxiv.org/pdf/2201.03726.pdf
> >
> > [3] Frey et al. Neural scaling of deep chemical models. https://chemrxiv.org/engage/api-gateway/chemrxiv/assets/orp/resource/item/627bddd544bdd532395fb4b5/original/neural-scaling-of-deep-chemical-models.pdf
> >
> > > 4. Equiformer architecture is very complicated, while there isn’t a clear motivation or intuition or explanation or thorough experimentation to support it. If the embedding module and the parameterization of DTP weights are considered, Equiformer could be more complicated.
> >
> > Thank you for pointing out this.
> >
> > We would like to clarify that **all the components have their motivation, intuition or reason as below**.
> > 1. Equiformer just follows the typical pre-norm architecture of Transformers. For example, they place layer normalization before attention or feed-forward networks and have skip connections. The only difference is that we use equivariant versions of operations as features can include scalars, vectors and other geometric tensors.
> >
> > 2. Each component in equivariant graph attention (Figure 1) has its intuition as follows.
> >
> > * (a) First, we combine features $x_i$ and $x_j$ in source and target nodes with linear layers and tensor products to obtain $f_{ij}$. $f_{ij}$ contains the information in the two nodes and considers geometric information by using tensor products.
> >
> > * (b) Second, since attention weights $a_{ij}$ should be invariant, we only consider the scalar parts of $f_{ij}$ and use MLP to compute MLP attention weights.
> >
> > * (c) Third, we want the feature sent from source node to target node to be non-linear, and therefore we apply gate activation to value vector $v_{ij}$. The reason that there is an additional DTP between $f_{ij}$ and $v_{ij}$ is that after gate activation, we want to mix non-linear features in different degrees.
> >
> > 3. For embedding, the node embedding is the same as other works, and the edge-degree embedding is used to encode the information of degrees at the beginning.
> >
> > 4. The parametrization of DTP is to make sure that when weights of DTP are generated from radial functions, the memory complexity is still manageable. Please see our response above for more details.
> >
> > If possible, can you please specify which part is confusing or not clear?

---

> > > ### Author Response · Authors · 2022-08-02
> > > **Response to Reviewer qLqa (3/3)**
> > >
> > > > 5. Section 3.1 (line 154-184) is less informative than Figure 2, while occupying more space and harder to understand. Figure 2 is more important than the verbose background introduction in Section 2 (Section 2.3 is good, while other parts could be more compact or omitted).
> > >
> > > Thanks for the valuable comments on the structure of the presentation.
> > >
> > > We would like to clarify that Section 3.1 provides all mathematical or implementation details for all operations in Equiformer.
> > >
> > > However, we agree with the reviewer that Figure 2 is informative and will add this to the main text and shorten Section 2.
> > >
> > > > 6. Given line 174 has already said $C_L$ is the number of channels for type-$L$ vectors, line 174-175 says “input $x$ containing $C_0 + \sum_{L=1}^{L_{max}} C_L$ type-0 vectors” is confusing. It might be better to replace this paragraph with Figure 2(c).
> > >
> > > Thanks for pointing out this.
> > >
> > > We would like to clarify that in Line 174, “$x$ containing $C_L$ type-L vectors with $0 < L <= L_{max}$” does not consider $L = 0$ and thus not type-$0$ vectors and should not be confusing as we already mention $L$ here is greater than $0$ and less than or equal to $L_{max}$. We will make this statement more clear and link the paragraph to Figure 2(c).
> > >
> > >
> > > > 7. All interactions between different type vectors and different elements within vector depend on DTP with SH(r_{ij}), while no experiment shows Self (Depth-wise) Tensor Product or Cross (Depth-wise) Tensor Product cannot improve the performance.
> > >
> > > Thanks for suggesting a potentially new operation.
> > >
> > > Please note that in our work, __we do not mention either self depth-wise tensor products or cross depth-wise tensor products.__
> > > If we understand correctly, the self depth-wise tensor product corresponds to taking tensor products of features at the same node.
> > > If this is what self tensor product means, we do not incorporate this type of operations in our network, and therefore we cannot conduct experiments.
> > > However, incorporating this operation can be an interesting future work.
> > >
> > >
> > > > 8. Comparison with Dot Product Attention could be more complete, e.g., key can be obtained from $x_j$  with a linear layer and attention bias can be obtained from the scalar part of $f_{ij}$ with a MLP before Reshape (splitting heads) or learnable radial function(mentioned in line 235-239) encoding $|| r_{ij} ||$, etc.
> > >
> > > It is our understanding that __we already consider similar cases in our comparison__.
> > >
> > > For dot product attention, the key is generated by a linear layer, a depth-wise tensor product and one final linear layer. If we omit the tensor product, there will be less information exchanged across different degrees, and this can potentially lead to worse performance and make the comparison less fair.
> > >
> > > For “attention bias obtained from the scalar part of $f_{ij}$ with MLP before Reshape”, this is exactly what we do to obtain MLP attention weights. The only difference is that here the reviewer suggests using the scalar part of both $f_{ij} ^ {(L)}$ and $f_{ij} ^ {(0)}$ in Figure 1 instead of only $f_{ij} ^ {(0)}$. However, note that $f_{ij} ^ {(L)}$ and $f_{ij} ^ {(0)}$ are obtained by applying linear layers (not tensor products) to the same feature and thus they contain similar information intuitively. Therefore, the comment suggests combining the proposed MLP attention with dot product attention and the original comparison is still fair. However, the combination can be an interesting future direction.
> > >
> > > For learnable radial functions, their effect is already included in the depth-wise tensor products (DTP) as the weights of DTP are parametrized by radial functions.
> > >
> > > __We believe that our comparison is already complete and fair as we only change MLP operation to dot product operation and leave other components the same.__
> > >
> > >
> > > > 9. The authors didn’t describe the limitations of their work.
> > >
> > > Please see our general response for clarification and additional limitations.

---

### Official Review · Reviewer_QY2z · 2022-07-11

**Rating:** 5
**Confidence:** 4
**Soundness:** 1 poor
**Presentation:** 3 good
**Contribution:** 2 fair

**Summary:**

This paper proposes an E(3)/SE(3) equivariant transformer on 3D molecular graphs. The central point is to apply a non-linear MLP attention mechanism based on the type-0 irreps features. Then the attention weights are multiplicated with other irreps features with type >0. The evaluations are carried out on QM9 and OC20, which somehow supports the benefit of the proposed method.

**Questions:**

1.	It is suggested to move the related work right after the introduction part, and provide more comparisons between different equivariant transformer methods.

2.	Line 107-118: does the type of inversion actually involved in the current implementation?

3.	Line 179-184: how to determine the pair (L, L’)? By random?

4.	Line 66-67: Not all GNNs are based on message passing based models.

5.	Line 254-258: the comparisons here are too shot to justify the contributions of this work.


**Limitations:**

The limitations and potential negative societal impact are not explicitly discussed.

**Strengths And Weaknesses:**

Strengths:

1.	This paper is overall compactly written and easy to follow.

2.	Besides QM9 (which is actually a well-explored dataset and could be overfitted by many recent methods), the authors evaluate their method on OC20, a new but more desirable dataset for performance comparison. It is valuable to see that the proposed method achieves good results on IS2RE. Necessary ablation studies are also performed.


Weaknesses:

1.	The biggest concern is that the methodological novelty is weak. It has been a common practice to explore transformers for graph data, such as Grover and Graphormer. And, even when considering equivariance, SE3-transformer and the works by [53,35] have investigated how to conduct attentions between equivariant features. While this paper does utilize certain slightly different components (such as LN, DTP), it is hard to see something essentially new in this submission compared to current papers. The authors claim the benefit of using MLP attention against the traditional dot-product counterpart. Yet, this replacement of attention computation just seems a trick other than a valuable contribution, particularly given that the MLP attention mechanism have already explored before in the graph learning field, such as GATv2 as cited by the authors.

2.	It is nice to see the evaluations on OC20. Yet, it seems Equiformer obtains better performance in terms of MAE, but not regarding EwT. Does this mean Equiformer is over-parametric and fits the data well in some cases, but not that good on average? Experimentally, it is still unknown why the proposed attention can help promote the performance, and why it is better than other equivariant transformers such as SE3-transformer and [53,35].

3.	SEGNN is a recent paper that can be regarded as a generalization version of many equivariant methods including tensor-field, LieConv, NequIP, EGNN, SE3 transformer, etc. Clearly, this paper has followed the notations and presentations in SEGNN (see for example the denotation in Eq. (6)). When compared to SEGNN, the equivariant attention in this paper actually degenerates to a specific case of SEGNN, for example, the attentions are given by Eq. (14) in SEGNN paper when \alpha(f_i, f_j) becomes \alpha(f_ij).

---

> ### Author Response · Authors · 2022-08-02
> **Response to Reviewer QY2z (1/2)**
>
> We thank the reviewer QY2z for the efforts and for acknowledging that __the evaluation supports the benefit of the proposed method and that it is valuable to achieve good results on OC20, a new but more desirable dataset for performance comparison, in addition to QM9 (which is actually a well-explored dataset and could be overfitted by many recent methods).__ We address the comments below.
>
>
> > 1. It has been a common practice to explore transformers for graph data, such as Grover and Graphormer. And, even when considering equivariance, SE3-transformer and the works by [53,35] have investigated how to conduct attentions between equivariant features. While this paper does utilize certain slightly different components (such as LN, DTP), it is hard to see something essentially new in this submission compared to current papers.
>
> Although attention has been applied to graph data, __the main question is whether those adaptations have room for improvement in the context of equivariant graph networks.__
>
> Compared to SE(3)-Transformer and works [53, 35], we improve upon it with the proposed novel and __more expressive__ and __general equivariant graph attention__ (MLP attention and non-linear message passing). __MLP attention and non-linear message passing are more expressive and improve upon dot product attention used in previous works on equivariant Transformers__ (Table 5 and Table 6). In contrast to works [53, 35], __we use a more general architecture capable of including higher degrees of tensors__ (L > 1). __Higher degrees of tensors can improve performance__ as shown by NequIP and SEGNN. To the best of our knowledge, we are among the first to propose this more expressive and general attention in the context of equivariant Transformers, and this is what differentiates this work from previous works.
>
> __We also compare this work with Graphormer__, which is mentioned by the reviewer (Table 4 in the main text and Table 7 and Table 8 in the appendix), and show that we can __achieve lower errors even with less training time__ (Line 626 - Line 629 in the appendix).
>
> Besides, the use of LN is to simply follow the practice of original Transformers and the use of depth-wise tensor product (DTP) is to make tensor products more efficient.
>
>
> > 2. The authors claim the benefit of using MLP attention against the traditional dot-product counterpart. Yet, this replacement of attention computation just seems a trick other than a valuable contribution, particularly given that the MLP attention mechanism have already explored before in the graph learning field, such as GATv2 as cited by the authors.
>
> We respectfully disagree with the statements.
>
> First, the proposed equivariant graph attention __consists of both MLP attention and non-linear message passing instead of merely MLP attention.__ Each of them is well justified in Table 5 and Table 6.
>
> Second, __MLP attention__ is explored in GATv2 but __its equivariant architecture has not yet been proposed and explored.__ __The combination of MLP attention and non-linear messages has not been explored either.__ Since the features in equivariant networks contain not only scalars but also vectors and other geometric tensors, it needs other operations and therefore careful modifications to use MLP attention. Moreover, our adaptation of MLP attention is general and can support tensors of higher degrees. In contrast, the dot product attention in works [53, 35] cannot support tensors of degrees higher than 1.
>
> Third, in our experiment, we find the case where MLP attention can clearly improve upon dot product attention (Table 6), suggesting that __the importance of MLP attention in the context of 3D atomistic graphs has not yet been well explored.__
>
>
> > 3. Yet, it seems Equiformer obtains better performance in terms of MAE, but not regarding EwT. Does this mean Equiformer is over-parametric and fits the data well in some cases, but not that good on average? Experimentally, it is still unknown why the proposed attention can help promote the performance, and why it is better than other equivariant transformers such as SE3-transformer and [53,35].
>
> We want to clarify that __Equiformer achieves the best mean absolute error (MAE) averaged over all sub-splits and second best energy within threshold (EwT) in Table 2 and Table 3.__
>
> Since Equiformer achieves the __lowest mean error__, it is __good on average__.
>
> However, the metric of EwT measures the percentage of predictions close enough to ground truth. Improving average errors would not always mean reducing errors of certain examples to below a threshold and would not always improve EwT metric. We have discussed the EwT metric in the appendix (Line 637 - Line 644).
>
> Detailed comparisons on equivariant networks based on irreducible representations (irreps) and equivariant Transformers can be found in our general response.

---

> > ### Author Response · Authors · 2022-08-02
> > **Response to Reviewer QY2z (2/2)**
> >
> > > 4. SEGNN is a recent paper that can be regarded as a generalization version of many equivariant methods including tensor-field, LieConv, NequIP, EGNN, SE3 transformer, etc. Clearly, this paper has followed the notations and presentations in SEGNN (see for example the denotation in Eq. (6)). When compared to SEGNN, the equivariant attention in this paper actually degenerates to a specific case of SEGNN, for example, the attentions are given by Eq. (14) in SEGNN paper when $\alpha(f_i, f_j)$ becomes $\alpha(f_{ij})$.
> >
> > We would like to mention that the authors of Tensor-Field Networks [1] and 3D Steerable CNNs [2], and Clebsch-Gordan Nets [3] proposed the framework of SE(3)-equivariant networks based on irreducible representations of SO(3) in 2018. A subset of these authors have developed the e3nn library [4], which NequIP, SEGNN and this work are based on.
> >
> > Equivariant graph attention is a more generalized version of message passing used in SEGNN. By setting the attention weights to be equal, it becomes SEGNN. However, for other cases, it is not SEGNN. Therefore, Equiformer can approximate SEGNN (attention weights being equal), but SEGNN (static weights) cannot approximate Equiformer (dynamic attention weights).
> >
> > Moreover, as shown in our experiment, Equiformer with both MLP attention and non-linear message passing can improve upon SEGNN (Table 1, Table 2, and Table 3).
> >
> > Reference:
> >
> > [1] Thomas et al. Tensor field networks: Rotation- and translation-equivariant neural networks for 3D point clouds. 2018.
> >
> > [2] Weiler et al. 3D Steerable CNNs: Learning Rotationally Equivariant Features in Volumetric Data. NeurIPS 2018.
> >
> > [3] Kondor et al. Clebsch-Gordan Nets: a Fully Fourier Space Spherical Convolutional Neural Network. NeurIPS 2018.
> >
> > [4] https://github.com/e3nn/e3nn
> >
> >
> > > 5. It is suggested to move the related work right after the introduction part, and provide more comparisons between different equivariant transformer methods.
> >
> > Thanks for the comment. We provide detailed comparisons to previous works in general response and will make the modification to the main text.
> >
> > > 6. Line 107-118: does the type of inversion actually involved in the current implementation?
> >
> > Yes, E(3) equivariance can be easily incorporated. We compared E(3) equivariance and SE(3) equivariance in Sec. D.1 and E.1 in the appendix and will make this more clear in the main text.
> >
> > > 7. Line 179-184: how to determine the pair (L, L’)? By random?
> >
> > L and L’ are determined based on the selection rules (i.e., Clebsch-Gordan coefficients) discussed in Line 127 to Line 129 (for SE(3)) and in Line 468 to Line 472 in the appendix (for E(3)). Additionally, L’ is chosen to be no greater than $L_{max}$ (Line 129 - Line 130).
> >
> > For example, given two inputs, the first input X containing two type-1 vectors (L = 1) and the second input Y containing one type-1 vector, the depthwise tensor product will result in two type-0 vectors, two type-1 vectors and two type-2 vectors. Thus, L’ can be either 0, 1, or 2. For each degree L’, the first output vector (first channel) depends only on the first vector in X (first channel) instead of all vectors (all channels) in X, and the second output vector (second channel) depends only on the second vector in X (second channel).
> >
> > Figure 2 and Figure 3 in the appendix illustrate the input and output features of depth-wise tensor products. The first input contains 2 type-0 vectors, 2 type-1 vectors and 1 type-2 vector, and the second contains 1 type-0 and 1 type-1 vectors.
> >
> > > 8. Line 66-67: Not all GNNs are based on message passing based models.
> >
> > Thanks for pointing out this.
> > We will mention “generally GNNs update features through message-passing layers” to take into account this fact.
> >
> > Can you please provide an example of performant GNNs not based on message passing?
> >
> > > 9. Line 254-258: the comparisons here are too shot to justify the contributions of this work.
> >
> > Thanks for pointing this out. We provide detailed comparisons to previous works in general response.
> >
> > > 10. Limitations and negative social impact.
> >
> > Please see our general response for clarification and additional limitations.
> >
> > As for negative social impact, we described one potential negative impact in Line 344 - Line 348, which is that the ability to accurately approximate quantum properties can be used for adversarial purposes like identifying, designing and creating hazardous chemicals.

---

> > > ### Comment · Reviewer_QY2z · 2022-08-07
> > > **Thanks for the response!**
> > >
> > > Greatly thank the authors for the detailed responses to my individual concerns and those general ones. The revised version is clearer. I am willing to raise the score to 5.
> > >
> > > However, I confirm that the technical contribution is still weak. I totally understand that MLP attention + non-linear message is the most desirable combination as supported by this paper. Extending current methods to the form proposed in this paper is not that surprising (the authors are unnecessarily suggested to further provide further explanations on this point). Hence, I can only suggest a borderline paper (tending to marginal acceptance) given the respectable experimental evaluations (QM-9, OC-20, MD-17) conducted by the authors.
> > >
> > > One more minor question: Is the attention only computed based the scalar features other than Irreps features?

---

> > > > ### Author Response · Authors · 2022-08-07
> > > > **Thanks for Update**
> > > >
> > > > We thank the reviewer for the response and update.
> > > >
> > > > For the question, here is the detailed answer.
> > > > We transform scalar features $f_{ij}^{(0)}$ in Figure 1 into attention weights with typical Leaky ReLU, one linear layer and softmax.
> > > > The scalar features $f_{ij}^{(0)}$ is, however, obtained by taking tensor products of irreps features ($x_{ij}$ and $SH(\vec{r_{ij}})$) and transforming with one typical linear layer as shown in Equation 3 in revision.

---

### Official Review · Reviewer_YZ7s · 2022-07-11

**Rating:** 6
**Confidence:** 4
**Soundness:** 3 good
**Presentation:** 3 good
**Contribution:** 4 excellent

**Summary:**

The paper proposes a transformer architecture which is equivariant to the Euclidean group SE(3). While other transformer architectures which are equivariant exists, the main novelty of this paper consists in the ability to incorporate higher dimensional irreps, thus, not limiting itself to the l=0,1 representations (scalars, vectors).

The authors also propose a novel attention mechanism which involves any dimensional irrep based on the tensor product. The authors test their models in the QM9, and OC20 datasets, and show that it achieves state-of-the-art performance in most tasks.


**Questions:**

It would have also been interesting to validate the model on some molecular dynamics datasets. For example, MD17 is nowadays a standard benchmarks for equivariant models.

Also, I think the relevant work section should belong in its entirety in the main text. I would suggest the authors to restructure the manuscript such that all the relevant information is present in the main text.


**Limitations:**


I wish the authors would discuss the limitation of their approach, for example whether the model becomes too parameter intensive for larger graphs.

Overall, I think the current paper is a nice work, and with a more extensive experimental section, involving also vectorial/tensorial tasks (like force prediction tasks in MD17) could qualify to be accepted in a venue like NeurIPS.

**Strengths And Weaknesses:**

The paper is clear and well written. The assumptions are clearly stated and the algorithm is clearly described. The paper addresses an interesting topic, that is, how to extend the inductive bias of equivariance to transformers.

The main weakness of the paper consists perhaps in the fact that the experimental section could have been more extensive and details. For instance, it would have been interesting to report the number of parameters for the various experiments as well as the training time, since it is often the case that transformers are quite heavy. Also, all the experiments have “scalar” tasks, and it would be very interesting how the model would perform for vectorial predictions.

---

> ### Author Response · Authors · 2022-08-02
> **Response to Reviewer YZ7s**
>
> We thank the reviewer YZ7s for the efforts and for acknowledging that __the paper is clear and well written, the assumptions are clearly stated, the algorithm is clearly described and that the work addresses an interesting topic.__ We address the comments below.
>
> > 1. It would have been interesting to report the number of parameters for the various experiments as well as the training time, since it is often the case that transformers are quite heavy.
>
> Please see our general response for training time and numbers of parameters.
>
> > 2. Also, all the experiments have “scalar” tasks, and it would be very interesting how the model would perform for vectorial predictions. MD17 is nowadays a standard benchmark for equivariant models.
>
> Please see our general response for results on MD17. Equiformer achieves better results than PaiNN and TorchMD-Net, which is also a previous work on equivariant Transformer.
>
> > 3. I would suggest the authors to restructure the manuscript such that all the relevant work information is present in the main text.
>
> Thank you for the suggestion on the related work section. We provide detailed comparisons to previous works in general response and will incorporate all the discussion in related work in the main text.
>
> > 4. I wish the authors would discuss the limitation of their approach, for example whether the model becomes too parameter intensive for larger graphs.
>
> Thank you for the comment. Please see our general response for clarification and additional limitations.
>
> As for scaling to larger graphs, since the memory complexity of Equiformer is dominated by pairs of nodes, Equiformer can theoretically scale better than models based on triplet or quadruplets representations like DimeNet [1] and GemNet-Q [2]. Besides, as mentioned by the work of Allegro [3], we can restrict the features exchanged within a local neighborhood in order to scale to larger graphs. The method proposed in Allegro is complementary but orthogonal to the proposed equivariant graph attention.
>
> Reference:
>
> [1] Gasteiger et al. Directional Message Passing for Molecular Graphs. ICLR 2020.
>
> [2] Gasteiger et al. GemNet: Universal Directional Graph Neural Networks for Molecules. NeurIPS 2021.
>
> [3] Musaelian et al. Learning Local Equivariant Representations for Large-Scale Atomistic Dynamics. 2022.

---

> > ### Comment · Reviewer_YZ7s · 2022-08-09
> > **answer**
> >
> >
> > I thank the authors for their responses, and I look forward to the discussion with the other reviewers.
> >
> > P.S. I understand the pressure to try to achieve an higher score, but I am not a huge fan of tricks like repeating and highlighting in bold positive feedback.

---

> > > ### Author Response · Authors · 2022-08-09
> > > **Response to Reviewer YZ7s**
> > >
> > > We thank the reviewer for the response.
> > > Sorry for having the tricks.

---

### Author Response · Authors · 2022-08-02
**General Response (1/3)**

We address some common comments by reviewers below.

> 1. Training time.

__The training time of all models including those in ablation study was reported in the appendix in the supplementary material__ (Line 587 ~ Line 591 for QM9 and Line 619 ~ Line 629 for OC20).

Particularly, we emphasize that for OC20 IS2RE + IS2RS, __Equiformer achieves less errors on the testing set__ (Table 7 and Table 8 in the appendix) and __takes 2.33X less training time compared to GNS + Noisy Nodes__ [1] and __15.5X less training time compared to Graphormer__ [2, 3] (champion of OC20 challenge in 2021). Note that as shown by Noisy Nodes [1], __under this setting, greater depths and more computation do translate to better performance and that Equiformer achieves better results with less computation.__

Reference:

[1] Godwin et al. Simple GNN Regularisation for 3D Molecular Property Prediction & Beyond. ICLR 2022.

[2] Ying et al. Do Transformers Really Perform Badly for Graph Representation? NeurIPS 2021.

[3] Shi et al. Benchmarking Graphormer on Large-Scale Molecular Modeling Datasets. 2022.


> 2. The number of parameters.


* QM9:
1. Equiformer (Table 1 and index 1 in Table 5): 3.53M.

2. Equiformer with only MLP attention (index 2 in Table 5): 3.01M.

3. Equiformer with only dot product attention (index 3 in Table 5): 3.35M.

4. E(3)-Equiformer (Table 9 in the appendix): 3.28M.

* OC20:
1. Equiformer (Table 2 and Table 3 and index 1 in Table 6): 9.12M.

2. Equiformer for IS2RE + IS2RS (Table 4 in the main text and Table 7 and Table 8 in the appendix): 26.8M.

3. Equiformer with only MLP attention (index 2 in Table 6): 7.84M.

4. Equiformer with only dot product attention (index 3 in Table 6): 8.72M.

5. E(3)-Equiformer (Table 11 in the appendix): 8.77M.


> 3. MD17 results.

We conduct experiments on MD17 following the setting of TorchMD-Net [1] and mainly compare with PaiNN [2] and TorchMD-Net. The numbers are mean absolute error of energy  (kcal/mol) and force (kcal/mol/Angstrom) predictions. The numbers of related works are taken from TorchMD-Net. As shown in the table, __Equiformer achieves overall better results on the MD17 dataset.__ Compared to PaiNN [2], Equiformer achieves better results on energy and force predictions of all molecules. Compared to TorchMD-Net [1], Equiformer achieves better energy predictions and significantly better force predictions for the first four molecules. For the last four molecules, Equiformer achieves higher energy errors but significantly lower force errors. We note that this is because of the ratio of force loss to energy loss and that further tuning the ratio could ensure lower energy and force errors.

Reference:

[1] Thölke et al. Equivariant Transformers for Neural Network based Molecular Potentials. ICLR 2022.

[2] Schutt et al. Equivariant message passing for the prediction of tensorial properties and molecular spectra. ICML 2021.

|     Molecule    |        |   | PaiNN | NequIP | TorchMD-Net | Equiformer |
|:---------------:|:------:|---|:-----:|:------:|:-----------:|:----------:|
|     Aspirin     | energy |   | 0.167 |        |       0.123 |      **0.122** |
|                 | force  |   | 0.338 |  0.348 |       0.253 |      **0.167** |
|     Benzene     | energy |   |       |        |       0.058 |      **0.051** |
|                 | force  |   |       |  0.187 |       0.196 |      **0.151** |
|     Ethanol     | energy |   | 0.064 |        |       0.052 |     **0.051** |
|                 | force  |   | 0.224 |  0.208 |       0.109 |      **0.071** |
| Malondialdehyde | energy |   | 0.091 |        |       0.077 |      **0.075** |
|                 | force  |   | 0.319 |  0.337 |       0.169 |      **0.133** |
|   Naphthalene   | energy |   | 0.116 |        |       **0.085** |      0.089 |
|                 | force  |   | 0.077 |  0.097 |       0.061 |      **0.044** |
|  Salicylic Acid | energy |   | 0.116 |        |       **0.093** |      0.101 |
|                 | force  |   | 0.195 |  0.238 |       0.129 |      **0.096** |
|     Toluene     | energy |   | 0.095 |        |       **0.074** |      0.084 |
|                 | force  |   | 0.094 |  0.101 |       0.067 |      **0.049** |
|      Uracil     | energy |   | 0.106 |        |       **0.095** |      0.099 |
|                 | force  |   | 0.139 |  0.173 |       0.095 |      **0.077** |

---

> ### Author Response · Authors · 2022-08-05
> **Update MD17 Results**
>
> We update our MD17 results by tuning the ratio of energy loss to force loss for each molecule and summarize the results below.
> We note that __Equiformer achieves better results for all molecules__.
>
> |     Molecule    |        |   | PaiNN | NequIP | TorchMD-Net | Equiformer |
> |:---------------:|:------:|---|:-----:|:------:|:-----------:|:----------:|
> |     Aspirin     | energy |   | 0.167 |        |       0.123 |      **0.122** |
> |                 | force  |   | 0.338 |  0.348 |       0.253 |      **0.167** |
> |     Benzene     | energy |   |       |        |       0.058 |      **0.051** |
> |                 | force  |   |       |  0.187 |       0.196 |      **0.151** |
> |     Ethanol     | energy |   | 0.064 |        |       0.052 |      **0.051** |
> |                 | force  |   | 0.224 |  0.208 |       0.109 |      **0.071** |
> | Malondialdehyde | energy |   | 0.091 |        |       0.077 |      **0.075** |
> |                 | force  |   | 0.319 |  0.337 |       0.169 |      **0.133** |
> |   Naphthalene   | energy |   | 0.116 |        |       **0.085** |      **0.085** |
> |                 | force  |   | 0.077 |  0.097 |       0.061 |      **0.048** |
> |  Salicylic Acid | energy |   | 0.116 |        |       0.093 |      **0.092** |
> |                 | force  |   | 0.195 |  0.238 |       0.129 |      **0.122** |
> |     Toluene     | energy |   | 0.095 |        |       **0.074** |      **0.074** |
> |                 | force  |   | 0.094 |  0.101 |       0.067 |      **0.056** |
> |      Uracil     | energy |   | 0.106 |        |       **0.095** |      0.096 |
> |                 | force  |   | 0.139 |  0.173 |       0.095 |      **0.086** |

---

### Author Response · Authors · 2022-08-02
**General Response (2/3)**

We address additional common comments here.

> 4. Limitations.

Although we describe one limitation in the experiment section, which is that MLP attention does not always improve upon dot product attention obviously (Line 322 - Line 324), we describe all potential limitations below and will include them in the appendix:

1. __Equiformer is based on irreducible representations (irreps) and therefore can inherit the limitations common to all equivariant networks based on irreps and the library e3nn [1].__ For example, using higher degrees L can result in larger features and using tensor products can be computationally intensive. Part of the reasons that tensor products can be computationally expensive are that the kernels have not been heavily optimized and customized as other operations in common libraries like PyTorch. But this is the issue related to software, not the design of networks. While tensor products of irreps naively do not scale well, if all possible interactions and paths (Line 132) are considered, some paths in tensor products can also be "pruned" for computational efficiency. We leave these potential efficiency gains to future work and in this work focus on general equivariant attention if all possible paths (up to $L_{max}$) in tensor products are allowed.

2. As we describe in the experiment section, __the limitation of the proposed attention is that the improvement can depend on tasks and datasets.__ For QM9, MLP attention improves not significantly upon dot product attention (Table 5). We surmise that this is because QM9 contains less atoms and less diverse atom types and therefore linear attention is enough. For OC20, MLP attention clearly improves upon dot product attention (Table 6). Non-linear messages improve upon linear ones for the two datasets.

3. __Equivariant graph attention requires more computation than typical graph convolution.__ It includes softmax operation and thus requires one additional sum aggregation compared to typical message passing. For non-linear message passing, it increases the number of tensor products from one to two and requires more computation. Along with 2., we note that if there is a constraint on training budget, using stronger attention (MLP attention and non-linear messages) would not always be optimal because for some tasks or datasets, the improvement is not obvious and using stronger attention can slow down training. For example, for the task of $C_{\nu}$ on QM9, using linear or non-linear messages results in the same performance (index 1 and index 2 in Table 6). However, non-linear messages increase the training time of one epoch from 6.6 minutes to 11 minutes (Line 587 - Line 591 in the appendix).

4. The proposed MLP attention has complexity proportional to the products of numbers of channels and numbers of edges. In the context of 3D atomistic graphs, the complexity is the same as that of messages and graph convolutions. However, in other domains like computer vision, the memory complexity of convolution is proportional to the number of pixels (nodes), not that of edges. Therefore, it would require further modification in order to use the proposed attention in other domains.

However, the attention used in Equiformer is restricted to local neighborhoods (e.g., within a pre-defined cutoff radius (see Table 10 and Table 12 in the appendix)). Therefore, the memory complexity of the proposed attention is the same as typical graph convolutions and is proportional to the number of edges, not the square of the number of nodes. Besides, we also show that using MLP attention is more computationally efficient than dot product attention in our network and achieves equal or better results.

[1] https://github.com/e3nn/e3nn

---

### Author Response · Authors · 2022-08-02
**General Response (3/3)**

> 5. Detailed comparison to equivariant networks based on irreducible representations (irreps) and equivariant Transformers.

Although __we have mentioned the differences between this work and other equivariant models based on irreps and other equivariant Transformers__ (Line 250 - Line 258 in the main text), we provide more details on the comparisons and some analysis here and will update the main text to be more clear:

#### __Equivariant networks based on irreducible representations (irreps)__:

1. __Tensor Field Networks (TFN)__ [1] and __NequIP__ [2] use only linear messages without attention (Line 252 - Line 253). NequIP additionally uses node-wise gate activation derived from 3D Steerable CNNs [3], and the gate activation is the same as this work.

2. __SEGNN__ [4] follows the practice of irreps and proposes to use non-linear messages (Line 256 - Line 257). The non-linear messages use gate activation mentioned above, and the difference from (a) is the additional usage of activation on edge features in addition to node features. As shown in their work, non-linear messages improve upon linear messages.

3. __Equiformer__ combines non-linear messages with non-linear MLP attention (Line 257 - Line 258) and the combination is better than either pure non-linear messages (compared to SEGNN in Table 1, Table 2 and Table 3) or pure MLP or dot product attention (Table 5 and Table 6). We note that MLP attention provides input-dependent attention weights and therefore using MLP attention along with non-linear messages can be more expressive than pure non-linear messages.

4. __Summary:__ The ranking of expressivity is that:
MLP attention + non-linear message (Equiformer) > non-linear message (SEGNN) > linear message (TFN and NequIP). This explains why Equiformer can be advantageous over previous models based on irreps.

#### __Equivariant Transformers__:

1. __SE(3)-Transformer__ [5] uses dot product attention with linear messages (Line 253 - Line 254) and the attention can support tensors of any degree L (e.g., L = 2).

2. __TorchMD-Net__ [6] and __EQGAT__ [7] also use dot product attention with linear messages. However, they design a more specialized architecture and the network can only use L = 0 and L = 1 tensors (Line 254 - Line 256).

3. __Equiformer__ uses non-linear MLP attention with non-linear messages (Line 12 - Line 14). As shown in Table 5 and Table 6 and discussed in Line 208 - Line 209 and Line 214 - Line 219, MLP attention improves upon dot product attention and non-linear messages improve upon linear messages. Therefore, __the proposed attention is more expressive than the attention used in previous equivariant Transformers.__ Moreover, __the proposed attention is general and can support tensors of any degree L, and using higher degrees L__ (since we use irreps) __can lead to better performance as shown in NequIP and SEGNN.__

4. __Summary:__ The proposed attention (MLP attention and non-linear messages) is more expressive than the attention (dot product attention and linear messages) used in all previous equivariant Transformers. The proposed attention is more general than TorchMD-Net and EQGAT and can support higher degrees L. They are the differences and advantages of Equiformer.

Reference:

[1] Thomas et al. Tensor field networks: Rotation- and translation-equivariant neural networks for 3D point clouds. 2018.

[2] Batzner et al. E(3)-equivariant graph neural networks for data-efficient and accurate interatomic potentials. Nature Communications 2022.

[3] Weiler et al. 3D Steerable CNNs: Learning Rotationally Equivariant Features in Volumetric Data. NeurIPS 2018.

[4] Brandstetter et al. Geometric and Physical Quantities improve E(3) Equivariant Message Passing. ICLR 2022.

[5] Fuchs et al. SE(3)-Transformers: 3D Roto-Translation Equivariant Attention Networks. NeurIPS 2020.

[6] Thölke et al. Equivariant Transformers for Neural Network based Molecular Potentials. ICLR 2022.

[7] Le et al. Equivariant graph attention networks for molecular property prediction. 2022.

---

### Author Response · Authors · 2022-08-07
**Upload Revision**

We thank reviewers for valuable comments on the presentation of the work and have updated our paper.

The differences are in blue and are summarized below:
1. We move the results in appendix (Table 7 and Table 8 in appendix) to the main text (Table 4 and Table 5 in revision).
2. We add more details (the content is the same as mentioned in our response) on related works and move the section of related works right after introduction. Due to the limited space, parts of related works are still in appendix, but we will move them to the main text if given more space.
3. We add a section of limitations in appendix (Section F).
4. We simplify the section of background and remove the parts of Graph Neural Networks and E(3)-Equivariant Neural Networks.
5. We move Figure 2 to the main text as suggested by reviewer qLqa and make some parts more clear based on reviewers' comments.
6. We add a subsection of discussion on how some components affect computational complexity (Section C.4).
7. We move appendix from supplementary material to the main text (revision).


Besides, we believe we have addressed reviewers' comments.
Please let us know if you have other questions or comments.

---

### Meta-Review · Area_Chair_EBE1 · 2022-08-28

**Recommendation:** Reject
**Confidence:** Less certain

**Metareview:**

This paper proposes Equiformer networks for predicting quantum properties based on 3D atomistic graphs. At the outset of the discussion period, the paper's scores were decidedly below borderline and the reviewers were concerned (i) that the methodological contribution of the paper was thin and (ii) about weaknesses in the experiments. Over the course of the discussion period, the authors engaged vigorously, providing additional experiments and moving several reviewers to increase their scores. However, at the resolution of the discussion period, despite the increases in score the paper remains overall below borderline. In general, the authors were more convinced by the experiments but still had misgivings that the technical contribution was thin and were even unsure about what precisely the technical contribution was in light of the massive related literature.

**Award:**

No

---

### Decision · Program_Chairs · 2022-09-14

Reject